# Flufenamic Acid-Loaded Electrospun Nanofibers Based on Chitosan/Poly(vinyl alcohol) Polymeric Composites for Drug Delivery in Biomedical Applications

**DOI:** 10.3390/polym17101411

**Published:** 2025-05-20

**Authors:** Kuppu Sakthi Velu, Mohammad Aslam, Ramachandran Srinivasan, Prathap Somu, Sonaimuthu Mohandoss

**Affiliations:** 1School of Chemical Engineering, Yeungnam University, Gyeongsan 38541, Republic of Korea; sakthi.velu4@yu.ac.kr (K.S.V.); mohammadaslam13@yu.ac.kr (M.A.); 2Centre for Ocean Research, Sathyabama Research Park, Sathyabama Institute of Science and Technology, Chennai 600119, Tamil Nadu, India; srinivasan.cor@sathyabama.ac.in; 3Department of Biotechnology and Chemical Engineering, School of Engineering, Faculty of Science, Technology and Architecture (FoSTA), Manipal University Jaipur, Dehmi Kalan, Off. Jaipur-Ajmer Expressway, Jaipur 303007, Rajasthan, India; 4Centre of Molecular Medicine and Diagnostics, Saveetha Dental College and Hospitals, Saveetha Institute of Medical and Technical Sciences, Saveetha University, Chennai 600077, Tamil Nadu, India

**Keywords:** chitosan, poly(vinyl alcohol), electrospinning, drug-release, antibacterial, anticancer, antioxidant

## Abstract

Nanostructured drug-delivery systems with enhanced therapeutic potential have gained attention in biomedical applications. Here, flufenamic acid (FFA)-loaded chitosan/poly(vinyl alcohol) (CHS/PVA; CSPA)-based electrospun nanofibers were fabricated and characterized for antibacterial, anticancer, and antioxidant activities. The FFA-loaded CSPA (FCSPA) nanofibers were characterized by scanning electron microscopy, Fourier-transform infrared spectroscopy, X-ray diffraction (XRD), and differential scanning calorimetry to evaluate their formation process, functional group interactions, and crystallinity. Notably, the average diameter of FCSPA nanofibers decreased with increasing CSPA contents (CSPA-1 to CSPA-3), indicating that FFA addition to CSPA-3 significantly decreased its diameter. Additionally, XRD confirmed the dispersion of FFA within the CSPA amorphous matrix, enhancing drug stability. FCSPA nanofibers exhibited a high swelling ratio (significantly higher than that of the CSPA samples). Biodegradation studies revealed that FCSPA exhibited accelerated weight loss after 72 h, indicating its improved degradation compared with those of other formulations. Furthermore, it exhibited a significantly high drug-encapsulation efficiency, ensuring sustained release. FCSPA nanofibers exhibited excellent antibacterial activity, inhibiting *Staphylococcus aureus* and *Escherichia coli*. Regarding anticancer activity, FCSPA decreased HCT-116 cell viability, highlighting its controlled drug-delivery potential. Moreover, FCSPA demonstrated superior antioxidation, scavenging DPPH free radicals. These findings highlight FCSPA nanofibers as multifunctional platforms with wound-healing, drug-delivery, and tissue-engineering potential.

## 1. Introduction

Electrospinning is a powerful and adaptable technology for fabricating nano-to-micrometer-scale fibers. It involves the alteration of various operational, process, and formulation parameters [1]. The fabrication of fibers through electrospinning of polymer solutions has been widely studied, with key parameters including applied voltage, tip-to-collector distance, solution feed rate, and solution characteristics [2]. The resultant fiber mats can exhibit a nonwoven and/or aligned fiber structure, depending on the type of collector used. Electrospinning has been used to fabricate nanofibers with extraordinary properties, such as extremely high surface area/volume ratios and high porosity owing to extremely small pore sizes. However, electrospun drug-carrying fibers exhibit high surface area/volume ratios, making them suitable for diffusion- and disintegration-based drug delivery [3]. Furthermore, electrospun nanofibers can encapsulate considerable amounts of drugs (up to 40%) and facilitate prolonged, multiple-phase, or rapid drug delivery [4]. Consequently, nonwoven mats comprising electrospun nanofibers have considerable potential for several uses, including biomedical applications such as drug delivery, wound dressing, and antibacterial uses [5].

Chitosan (CHS), a (1-4)-linked 2-amino-2-deoxy-D-glucopyranose, is a derivative of chitin, which is a common naturally occurring polysaccharide [6]. Because of its biocompatible, renewable, biodegradable, wound-healing, antimicrobial, and antitumor properties, CHS has attracted significant interest for its application in wound dressing, wound healing, drug delivery, and tissue engineering [7]. Therefore, CHS-based nanofibers have emerged as valuable biological materials owing to their biocompatibility, biodegradability, and antibacterial characteristics. However, the amine (NH_2_) group in the backbone of CHS saccharide and the strong hydrogen bonding between NH_2_ and hydroxyl (–OH) groups prevent continuous fiber formation in this material [8]. However, its polycationic behavior in solution limits the feasibility of directly synthesizing CHS through electrospinning [9]. Numerous studies have attempted to increase the electrospinning capabilities of CHS by combining it with other polymers, such as poly(vinyl pyrrolidone) [10], poly(vinyl alcohol) (PVA) [11], poly(ethylene oxide) [12], silk fibroin [13], and zein [14].

Among them, PVA is a synthetic, linear, semicrystalline polymer composed of a carbon chain and an OH group as its backbone and functional component, respectively [15,16]. PVA exhibits numerous essential characteristics, such as easy accessibility, water solubility, high film-forming capacity, and thermostability [17]. Furthermore, PVA exhibits superior fiber-forming capacity and hydrophilicity, and fibers electrospun from PVA have been commercialized since the 1950s [18]. Furthermore, PVA can improve fiber spinning by reducing the repulsive forces in charged polymer solutions. Notably, CHS can be integrated with PVA for improved electrospinning, considering their relative miscibility and comparable electrospinning behaviors, e.g., coagulation, orientation, and cross-linking [18]. Integrating PVA and CHS imparts the solution with the features of each component and with additional valuable properties, such as good electrospinnability and mechanical capabilities. Thus, CHS/PVA (CSPA) is a candidate for antimicrobial applications [19]. Additionally, electrospun CSPA nanofibers have been extensively investigated in recent years because of their biocompatibility, biodegradability, and antibacterial capabilities in tissue engineering [20,21,22,23].

Several nano-delivery systems, including nanofibers, nanoparticles, nanocapsules, nanoliposomes, and dendrimers, have been investigated for antibacterial drug delivery [24]. The encapsulation of drugs in electrospun nanofibers offers several advantages, including high surface area/volume ratios and porosities [25,26]. Compared to other nanocarriers used for antibacterial drug delivery, nanofibers demonstrate high drug-loading capacity, excellent encapsulation efficiency (EE%), minimal systemic toxicity, and enable both sustained and controlled release profiles. However, no study has experimentally attempted the electrospinning-based loading of CSPA with flufenamic acid (FFA), i.e., the electrospinning-based preparation of FFA-loaded CSPA (FCSPA). FFA, a non-steroidal anti-inflammatory drug, has garnered attention for its drug-delivery potential as well as for its antibacterial, cytotoxic, and anticancer applications [27,28]. FFA exhibits remarkable antibacterial and antibiofilm properties; it significantly interrupts biofilm formation and limits bacterial growth, making it a good alternative for fighting resistant diseases [29,30]. Furthermore, assessing the cytotoxic and anticancer characteristics of this composite structure may provide new avenues for drug-delivery systems targeting cancer cells [27,31].

In the present study, FFA-loaded CHS/PVA nanofibers were prepared by electrospinning, and the properties of CHS- and PVA-based composite nanofibers in different ratios were investigated. The prepared CSPA and FFA-loaded CSPA (FCSPA) nanofibers were characterized by scanning electron microscopy (SEM), Fourier-transform infrared (FTIR) spectroscopy, X-ray diffraction (XRD), and differential scanning calorimetry (DSC) to evaluate their surface morphologies, functional group contents, surface areas, and thermal properties, respectively. Furthermore, the effects of various CSPA ratios on the swelling ratio, in vitro biodegradation, EE%, and release behavior of FCSPA nanofibers were investigated. The antibacterial activities of CSPA and FCSPA nanofibers against gram-negative *Escherichia coli* and gram-positive *Staphylococcus aureus* were determined by colony counting. Additionally, the anticancer and antioxidant activities of CSPA and FCSPA were extensively investigated. Our findings demonstrate that FCSPA nanofibers are versatile platforms with potential for wound healing, drug delivery, and tissue engineering.

## 2. Materials and Methods

### 2.1. Materials

CHS (molecular weight [MW]: 50–190 kDa; 95% deacetylation degree) and FFA were obtained from Sigma-Aldrich (St. Louis, MO, USA). PVA (MW: 146–186 kDa, 89% deacetylation degree) was procured from Daejung Company Ltd., Korea (Busan, Republic of Korea), and deionized (DI) water was used as the solvent for the solutions. All chemicals and solvents were used without further purification. The human colon cancer cell line (HCT-116) was purchased from the Korean Cell Line Bank (Seoul, Republic of Korea).

### 2.2. Preparations of Chitosan/Poly(vinyl alcohol) and Flufenamic Acid-Loaded Chitosan/Poly(vinyl alcohol) Solutions

To prepare the CSPA and FCSPA polymer solutions, 0.3 g of CHS (3% *w*/*v*) was dissolved in 10 mL of 1% (*v*/*v*) acetic acid and stirred at ambient temperature for 3 h until a clear solution was obtained. Concurrently, 2 g of PVA (10% *w*/*v*) was dissolved in 20 mL of distilled water and stirred at 50 °C using a magnetic heater-stirrer for 4 h to ensure complete dissolution. Thereafter, the CHS and PVA solutions were mixed at different weight ratios (*w*/*w*): 30/70, 50/50, and 70/30 for CSPA-1, CSPA-2, and CSPA-3, respectively, followed by stirring at ambient temperature for 1 h to achieve homogeneity (Table 1). For the FCSPA solutions, the optimized CSPA sample, i.e., CSPA-3 ratio (70/30 *w*/*w*), was selected. FFA was dissolved in the mixing solution at a concentration of 25 mg in 5 mL (Table 1). Subsequently, the mixture was continuously stirred until full dissolution was achieved, ensuring uniform dispersion of all components.

### 2.3. Electrospinning Procedure

The prepared CSPA-1, CSPA-2, CSPA-3, and FCSPA solutions were loaded into a 5 mL syringe with a 22 G needle and secured using a syringe pump. The flow rate was set to 1 mL/h, with a maintained needle-collector distance of 14 cm. Next, they were subjected to a high voltage of 30 kV, and the room humidity was controlled at 60% to facilitate uniform fiber formation (Figure 1). Thereafter, the electrospun fibers were collected on an aluminum foil-wrapped collector, washed multiple times with distilled water to remove the residual acetic acid, and dried overnight in a desiccator at room temperature.

### 2.4. Characterizations

Before the electrospinning procedure, the shear viscosities at 100 s^−1^; solution conductivities; and surface tensions of the CSPA-1, CSPA-2, CSPA-3, and FCSPA solutions were tested. Their viscosities were analyzed using a Brookfield LVT viscometer with a small-sample thermostated adapter, spindle, and chamber SC4-18/13R at 25 ± 0.1 °C. Further, their conductive characteristics were measured using an Orion 162 conductivity meter at room temperature, and their surface tensions were measured using the pendant drop method and a tensiometer (OCA20, Dataphysics Instruments, Filderstadt, Germany). For all in vitro assays, CSPA-1, CSPA-2, CSPA-3, and FCSPA nanofiber suspensions were prepared at a standardized concentration (10 mg) per well to ensure consistent exposure and account for the dose-dependent nature of the observed biological responses.

#### 2.4.1. Scanning Electron Microscopy

The surface morphologies and fiber structures of the CSPA-1, CSPA-2, CSPA-3, and FCSPA electrospun nanofibers were analyzed by SEM (FESEM JSM-7600F, JEOL, Tokyo, Japan). The samples were sputter-coated with gold using an auto fine coater before imaging to enhance their conductivities. SEM imaging was performed at an accelerating voltage of 200 kV, and the fiber diameters were measured using ImageJ software (1.54j); 50 fiber diameters were randomly selected for statistical analyses.

#### 2.4.2. Fourier-Transform Infrared Spectroscopy

FTIR spectroscopy was performed to determine the chemical functional groups in the samples and to examine the molecular interactions in the pure compounds, i.e., CSPA-3, and FCSPA nanofibers. The FTIR spectra were verified within the 4000–400 cm^−1^ range using an attenuated total reflectance module–equipped FTIR spectrometer (Perkin-Elmer, Waltham, MA, USA).

#### 2.4.3. X-Ray Diffraction

XRD (Shimadzu Corporation, Kyoto, Japan) was performed to investigate the crystalline structures of the pure molecules, CSPA-3, and FCSPA nanofibers. The measurements were performed using a Cu–Kα radiation source at 40 kV in a 2θ range of 10–80°.

### 2.5. Swelling Ratio

The swelling properties of the CSPA-1, CSPA-2, CSPA-3, and FCSPA nanofibers were evaluated. To achieve this, the samples were immersed in phosphate-buffered saline (PBS) solution at pH 7.4 and room temperature. The electrospun CSPA-1, CSPA-2, CSPA-3, and FCSPA nanofibers were cut into uniform pieces (1 cm × 1 cm) to ensure consistency during testing. The initial dry weight (W_0_) of each sample was recorded using a high-precision analytical balance. Subsequently, the samples were fully immersed in the PBS solution for predetermined intervals (0, 1, 2, 3, 4, 5, 6, 9, 12, 15, 18, 21, and 24 h) while ensuring complete submersion and minimal interference. At each interval, the samples were carefully removed using tweezers, and excess surface water was gently blotted using filter paper to prevent structural deformation. The swollen weight (W_1_) was immediately measured and recorded.

The swelling percentage was calculated as follows:Swelling ratio (%) = [(W_1_ − W_0_)/W_0_] × 100,(1)
where W_0_ is the initial dry weight and W_1_ is the swollen weight after immersion.

### 2.6. Degradation Profile

The in vitro degradation behaviors of the CSPA-1, CSPA-2, CSPA-3, and FCSPA nanofibers were evaluated by immersing them in PBS (pH 7.4) containing 0.15% (*w*/*v*) lysozyme to simulate physiological enzymatic degradation. The nanofiber mats (~10 mg) were cut into uniform pieces (1 cm × 1 cm) and placed in 10 mL PBS solution at 37 °C inside a shaking incubator set at 100 rpm. At specified intervals (24, 48, and 72 h), the samples were carefully removed, rinsed with DI water to remove the remaining salts, and dried in a vacuum at 40 °C until a stable weight was achieved.

The degradation rate was determined as follows:Degradation (%) = [(W_0_ − W_t_)/W_0_] × 100,(2)
where W_0_ is the initial dry weight and W_t_ is the remaining dry weight at each point.

### 2.7. Drug-Encapsulation Efficiency and Drug Release

To determine the drug EE% of the CSPA-3 electrospun nanofibers, we dissolved a determined mass of each sample in 1 mL of acetic acid. Thereafter, the FFA concentration of the solution was measured using high-performance liquid chromatography (Agilent Technology, Santa Clara, CA, USA) at a maximum wavelength (λ_max_) of 254 nm. Subsequently, we developed a conventional calibration curve for FFA, and the EE% was computed as follows:EE% = Total amount of encapsulated FFA/Theoretical amount of FFA × 100(3)

The release profile of FFA from the CSPA-3 nanofibers was evaluated in PBS (pH 7.4) at 37 °C under dynamic conditions. The electrospun FCSPA-3 nanofiber mats were cut into 1 cm × 1 cm sections and precisely weighed (~10 mg). The nanofiber mats were immersed in 50 mL PBS (pH 7.4) in sealed glass vials. These vials were incubated in a shaking water bath or an orbital shaker at 37 ± 0.5 °C with gentle agitation at 100 rpm. To maintain the sink conditions, 1 mL of the release medium was withdrawn and replaced with an equal volume of fresh PBS at predetermined intervals (0, 1, 2, 3, 4, 5, 6, 9, 12, 15, 18, 21, and 24 h). Thereafter, the obtained samples were filtered using 0.45 μm membrane filters, and the concentration of the released FFA was measured by ultraviolet–visible (UV–Vis) spectrophotometry at λ_max_ = 254 nm. The calibration curve for FFA in PBS was developed to assess the drug concentration, after which the cumulative drug release (%) was estimated.

### 2.8. Bacterial Culture

The antibacterial activities of the FFA, CSPA-1, CSPA-2, CSPA-3, and FCSPA nanofibers were assessed against *S. aureus* and *E. coli* using the colony-counting method. To begin, bacterial suspensions were prepared by inoculating fresh bacterial cultures in Mueller–Hinton broth (MHB) using an incubation time of 24 h at 37 °C until they reached a concentration of 10^6^ CFU/mL. The electrospun nanofiber samples (3.5 cm^2^) were sterilized under UV light for 30 min before immersion in separate tubes containing 10 mL of bacteria-inoculated MHB. The tubes were incubated at 37 °C with continuous shaking at 100 rpm, and at predetermined intervals (0, 2, 4, 6, 12, and 24 h), 100 µL aliquots of the bacterial suspension were withdrawn, serially diluted in sterile PBS, and spread onto Mueller–Hinton agar plates. After 24 h of incubation at 37 °C, the colony-forming units were counted and compared with those of the control group (ciprofloxacin and vancomycin), consisting of bacteria without nanofiber treatment. The percentage reduction in bacterial viability was subsequently calculated as follows:Bacterial reduction (%) = [(CFU_control − CFU_treated)/CFU_control] × 100(4)

The bacterial cultures (initial optical density at 600 nm [OD_600_]: ~0.05) were treated with FFA-loaded nanofibers, free FFA, or a standard antibiotic (vancomycin and ciprofloxacin) in 96-well plates. The cultures were incubated with shaking at 37 °C, and the OD_600_ was measured at 0, 4, 8, 12, and 24 h using a microplate reader. All treatments were performed in triplicate to evaluate the time-dependent antibacterial effects.

### 2.9. Cytotoxicity Assay

The cytotoxicities of the FFA, CSPA-1, CSPA-2, CSPA-3, and FCSPA nanofibers were assessed on HCT-116 cell lines using the 3-[4,5-dimethylthiazol-2-yl]-2,5 2,5-diphenyl tetrazolium bromide (MTT) assay. The HCT-116 cells were cultured in Dulbecco’s modified Eagle’s medium (DMEM) supplemented with 10% fetal bovine serum, 100 U/mL penicillin, and 100 μg/mL streptomycin and maintained at 37 °C in an incubator containing 5% CO_2_. Electrospun nanofiber samples were cut into 6 mm diameter discs, sterilized under UV light for 30 min, and placed in 96-well culture plates. The cells were seeded at a density of 5 × 10^3^ cells per well and incubated for 24 h to facilitate cell attachment. Following incubation, the culture medium was replaced with fresh medium containing FFA, CSPA-1, CSPA-2, CSPA-3, and FCSPA nanofibers, which were prepared via immersion in DMEM for 48 h at 37 °C. Next, 20 μL of MTT solution (5 mg/mL in PBS) was added to each well, followed by incubation for 4 h. Formazan crystals were immersed in 150 μL of dimethyl sulfoxide, and the absorbance was measured at 570 nm using a microplate reader. Thereafter, the cell viability was estimated using the following equation:Cell viability (%) = (Absorbance of test sample/Absorbance of control) × 100(5)

### 2.10. Cell Imaging

To investigate the cell adhesion and morphology of the samples, HCT-116 cells were seeded onto a control as well as onto FFA, CSPA-1, CSPA-2, CSPA-3, and FCSPA nanofiber scaffolds at a density of 5 × 10^3^ cells per well in 24-well plates. Next, the cells were incubated at 37 °C in a 5% CO_2_ environment for 48 h to facilitate attachment and proliferation. Subsequently, they were fixed with 4% paraformaldehyde for 15 min, washed three times with PBS, and permeated with 0.1% Triton X-100 for 5 min. Next, the nuclei were counterstained with 4′,6-diamidino-2-phenylindole (DAPI) and propidium iodide (PI) for 10 min. Following staining, the samples were rinsed with PBS and observed under a fluorescence microscope to assess cell morphology.

### 2.11. Antioxidant Activity

The antioxidant activities of free FFA, CSPA-1, CSPA-2, CSPA-3, and FCSPA nanofibers were determined using a 2,2-diphenyl-1-picrylhydrazyl (DPPH) free radical scavenging assay. Each 10 mg sample was separately immersed in 1 mL of DPPH ethanolic solution (10^−4^ mol/L) and incubated in the dark at room temperature for varied durations (6, 12, 18, and 24 h). Additionally, a DPPH solution containing FFA was used as the control. The percentage of DPPH-scavenging activity was calculated as follows:DPPH-scavenging activity (%) = A_0_ − A_i_/A_0_ × 100(6)
where A_0_ is the absorbance of the control solution (DPPH) and A_i_ is the absorbance of the solution containing the corresponding free FFA, CSPA-1, CSPA-2, CSPA-3, and FCSPA nanofibers.

### 2.12. Statistical Analysis

The studies were performed in triplicate, and the data were reported as mean ± standard deviation. SPSS version 27.0 was used to perform a one-way analysis of variance, followed by Tukey’s test. A *p*-value < 0.05 was considered statistically significant.

## 3. Results and Discussion

### 3.1. Nanofiber Morphologies

The surface morphologies and physical properties of the electrospun scaffolds were analyzed by SEM. Figure 1 shows the effects of varying the CSPA ratios (30/70 [CSPA-1], 50/50 [CSPA-2], and 70/30 [CSPA-3]), along with the FFA-loaded CSPA-3 fiber mats, designated as FCSPA. The SEM images revealed that the electrospun CSPA-1, CSPA-2, and CSPA-3 nanofibers were bead-free, continuous, and randomly oriented [32]. A comparative analysis of Figure 1a,c,e demonstrates that increasing the CHS content caused a substantial increase in the average diameter of the nanofiber, induced noticeable morphological alterations, and caused an increase in the random-alignment degree [32]. Specifically, at a CSPA ratio of 70/30 (*w*/*w*), the nanofibers appeared to be smooth and without bead defects (Figure 1e), making this composition optimal for further investigation. Among the electrospun nanofibers, the FCSPA variant exhibited reduced average fiber diameter with increasing CSPA content (Figure 1g) [33]. Notably, FFA incorporation caused a significant decrease in the fiber diameter as the size of the CSPA-3 nanofiber decreased from 347 ± 61 nm to 81 ± 27 nm, following FFA-drug loading (Figure 1i–l) [34]. The variations in the CSPA content affected the fiber diameter and indirectly affected the FFA-loading efficiency of the CSPA-3 nanofibers. Notably, CHS addition increased the average fiber diameter, as observed with the increasing CHS contents of the CSPA-1, CSPA-2, and CSPA-3 formulations. In contrast, the presence of FFA in the CSPA-3 formulation (FCSPA) caused a reduction in the average fiber diameter, likely due to the enhanced surface charge and electrostatic repulsion, which facilitated the formation of finer nanofibers.

### 3.2. Surface Properties of Nanofibers

The FTIR spectra of CHS, PVA, CSPA (*w*/*w*: 70/30), CSPA-3 nanofibers, FFA, FFA-loaded CSPA (*w*/*w*: 70/30), and FCSPA nanofibers are shown in Figure 2. In the FTIR spectrum of CHS, the peaks at 1649 and 1547 cm^−1^ corresponded to the C=O vibration and N–H bending, respectively (Figure 2a). The band at 1380 cm^−1^ was associated with CH_2_ deformation, and the absorption band at 1077 cm^−1^ corresponded to the stretching of the C–O–C bond [15]. The FTIR spectrum of PVA showed a broad O–H stretching band at 3200–3600 cm^−1^, indicating the presence of –OH groups and hydrogen bonding, whereas C–H stretching occurred at 2800–3000 cm^−1^, indicating methylene (–CH_2_) vibrations (Figure 2b). The C=O stretching at 1659 cm^−1^ corresponded to residual acetate groups, and the O–H bending at 1592 cm^−1^ was related to the absorbed water [16]. Furthermore, C–H bending was observed at 1400–1500 cm^−1^, and C–O stretching was observed around 1141 cm^−1^ and was associated with the polymer crystallinity. The FTIR spectrum of the CSPA-3 nanofibers showed broad O–H and N–H stretching peaks at 3278–3326 cm^−1^, indicating hydrogen bonding between –OH and NH_2_ (Figure 2c). Furthermore, C–H stretching was observed at 2800–2950 cm^−1^, corresponding to –CH_2_ vibrations, whereas the C=O stretching band at 1640–1690 cm^−1^ was attributed to the presence of polymer interactions or residual acetate groups [21]. The C–O–C and C–N stretching between 1085 and 1144 cm^−1^ reflected the polysaccharide structures of CHS and PVA.

A broad N–H stretching band was observed in the FTIR spectrum of FFA at 3321 cm^−1^, and it corresponded to the presence of a secondary NH_2_ group. The C=O stretching vibration at 1652 cm^−1^ corresponded to the carbonyl group, whereas the C–H stretching between 2800 and 3100 cm^−1^ corresponded to the aromatic and aliphatic C–H vibrations (Figure 2d). The C=C stretching at 1500–1600 cm^−1^ represented vibrations of the aromatic rings, and the C–F stretching between 1100 and 1400 cm^−1^ was attributed to the presence of a trifluoromethyl group [35]. The FTIR spectrum of FCSPA nanofibers exhibited broad O–H and N–H stretching bands at 3200–3500 cm^−1^, highlighting hydrogen bonding from –OH and NH_2_ (Figure 2e). The C=O stretching peak at 1650 cm^−1^ indicated FFA incorporation into the CSPA-3 matrix. The C–H stretching vibrations at 2800–3000 cm^−1^ were attributed to the CH_2_ groups, and the C–O–C and C–N stretching bands between 1085 and 1145 cm^−1^ indicated the formation of ether and NH_2_ bonds, representing the FCSPA nanofibers. The incorporation of FFA largely shifted the peak positions with changes in intensity, highlighting the interactions between the FFA drug and the CSPA-3 matrix. These spectral characteristics confirmed the integration of FFA into the CSPA-3 nanofiber matrix.

The XRD patterns of the crystalline and amorphous structures of pure CHS, PVA, CSPA (*w*/*w*; 70/30), CSPA-3 nanofibers, FFA, FFA-loaded CSPA (*w*/*w*; 70/30), and FCSPA nanofibers were analyzed (Figure 3). The XRD patterns of CHS and PVA exhibited broad, diffuse peaks. CHS exhibited a distinct peak at approximately 2θ = 20.3° (Figure 3a), whereas PVA exhibited a peak around 2θ = 19.2° (Figure 3b), characteristic of an amorphous phase [32,33]. In contrast, the XRD spectrum of the electrospun CSPA (*w*/*w*; 70/30) nanofibers (Figure 3c) exhibited decreased intensity for the peaks associated with CHS and PVA, indicating the formation of intermolecular and intramolecular hydrogen bonds between CHS and PVA during electrospinning [32,33]. The XRD pattern of FFA revealed sharp, well-defined peaks at 2θ of 13.4°, 14.6°, 17.8°, 18.3°, 19.2°, 22.7°, 24.3°, 26.1°, 30.1°, and 31.0°, indicating its crystalline nature (Figure 3d) [36]. However, the XRD pattern of the FFA-loaded CSPA (*w*/*w*: 70/30) nanofibers did not show the sharp crystalline peaks of FFA. The peaks were replaced by broad peaks. This change in the diffraction pattern indicates disruption of the crystalline structure of FFA owing to the dissolution of FFA in water and its blending with the CSPA matrix during electrospinning (Figure 3e). Consequently, the absence of distinct diffraction peaks for FFA may be attributed to its low concentration in the composite fibers, which probably falls below the detection threshold of the instrument. Additionally, FFA may be present in amorphous or molecularly dispersed states within the CSPA matrix, further contributing to the lack of identifiable crystalline features [37].

### 3.3. Thermal Properties of Nanofibers

Figure 4 depicts the DSC analyses of CHS, PVA, CSPA (*w*/*w*; 70/30), CSPA-3 nanofibers, FFA, FFA-loaded CSPA (*w*/*w*; 70/30), and FCSPA nanofibers. The DSC study of pure CHS and PVA revealed a rather large and highly endothermic curve with peaks at 196 °C and 192 °C (Figure 4) [20,22]. Moreover, the CSPA-3 nanofiber exhibited a peak shift toward lower temperatures, resulting in the development of a broad endothermic curve at 164 °C. The melting-point reduction of PVA in the blends indicated that there was slight mixing between PVA and CHS [22,23]. Furthermore, the decrease in the endothermic heat was attributable to the presence of amorphous CHS, which damaged the crystalline structure of PVA in the polymer blends. The pure FFA drug exhibited a pronounced endothermic peak at 160 °C, indicating crystal melting. Similarly, the peaks of the endothermic curve of FCSPA nanofibers migrated to higher temperatures at 212 °C, following the addition of FFA-loaded CSPA-3 nanofibers. Upon the addition of FFA, the endothermic curves of the FCSPA nanofibers became less noticeable and eventually disappeared. These results indicated substantial interactions between the functional groups of CSPA and FFA. The DSC results exhibited good agreement with the XRD pattern, indicating consistent thermal and structural properties of the material.

### 3.4. Swelling Evaluation

The water-absorption capacities of the scaffolds were evaluated by measuring their swelling in a PBS solution. The results demonstrated that all nanofibers facilitated effective water uptake, which increased with prolonged exposure to the aqueous environment. To determine the swelling percentages of the CSPA-1, CSPA-2, CSPA-3, and FCSPA nanofiber mats (Figure 5a), we determined their equilibrium swelling ratios at 24 h, obtaining approximately 169 ± 8.4%, 187 ± 6.3%, 245 ± 12.2%, and 302 ± 15.1%, respectively. As shown in Figure 5a, differences were observed in the swelling behaviors of the scaffolds, with the FCSPA nanofiber mats exhibiting the highest water uptake compared with those of CSPA-1, CSPA-2, and CSPA-3 [38]. The increase in the swelling percentage may be attributed to the enhancement of the –OH groups in the FCSPA nanofibers during blending. Additionally, the higher swelling capacity of the FCSPA nanofibers indicated that the presence of FFA in the scaffolds increased their surface area, making them more suitable for cell adhesion and infiltration. Overall, the FFA-loaded CSPA-3 nanofibers exhibited high absorption capacities (>300%), maintaining a stable swollen state [39]. This property is beneficial for preserving moisture conditions, making them promising for wound-healing applications as well as ensuring controlled and sustained drug release in drug-delivery systems. The SEM graphs (Figure 5b,c) show that the FCSPA nanofibers maintained their integrity after being immersed in PBS for 24 h, thereby indicating their stability in water solution.

### 3.5. In Vitro Degradation

Biodegradation is a key factor in determining the suitability of nanofibers for biomedical applications. The CSPA-1, CSPA-2, CSPA-3, and FCSPA nanofiber mats exhibited progressive weight loss over time (Figure 5d). Notably, the degradation rate of the FCSPA nanofiber mat was higher than that of CSPA-1, CSPA-2, and CSPA-3. This may be due to the incorporation of FFA into the CSPA-3 formulation, which disrupts polymer chain entanglement and accelerates the degradation process [40]. The highest degradation rate was observed in the FCSPA nanofibers, likely due to the increased content of the highly hydrophilic polymer PVA. After 72 h, the degradation rates for CSPA-1, CSPA-2, and CSPA-3 were 77.2 ± 2.38%, 79.0 ± 3.07%, and 83.6 ± 3.19%, respectively. In contrast, the FCSPA scaffolds exhibited significantly higher degradation rates (98.12 ± 3.31%), highlighting their excellent biodegradability [40]. The degradation behaviors of the FCSPA nanofibers before and after PBS immersion for 24 h were investigated by SEM. In Figure 5e, the morphology of the FCSPA nanofibers appeared to have degraded; however, their fibrous structure was still visible before 72 h. In contrast, Figure 5f shows that after 72 h, the surface leveled off, with some pores present on the surface. This change was attributed to the excellent water solubility of PVA in the blend, which ensured moisture retention and facilitated the biodegradation of the FCSPA nanofibers.

### 3.6. Loading Efficiency and In Vitro Drug Release

The absence of thermal reactions during electrospinning is advantageous for the encapsulation of thermolabile bioactive substances. As illustrated in Figure 6a, the EE% of FFA in CSPA-1, CSPA-2, and CSPA-3 nanofibers were 77.3 ± 1.86%, 83.4 ± 2.31%, and 91.2 ± 0.95%, respectively. In all FFA-loaded nanofibers, the EE% exceeded 75%, demonstrating the effective dispersion of FFA within the CSPA solution, where it remained stably encapsulated after electrospinning [41]. A notable increase in EE% was observed when the CSPA ratio was adjusted to 70/30 (*w*/*w*). This enhancement was attributed to the increased CHS concentration, which facilitated the formation of additional hydrogen bonds between CHS and PVA. Consequently, this interaction likely weakened the binding capacity of FFA with the CSPA matrix, resulting in an enhanced EE%.

UV–Vis spectroscopy was performed to analyze the release profile of FFA from the CSPA-3 nanofiber mats, providing insights into the structure–function relationship of the electrospun FCSPA nanofibers in PBS within 24 h (Figure 6b). The release behavior of free FFA was used as a control. After 24 h, the release rate of FFA from the FCSPA nanofibers reached 93.27 ± 2.31%, whereas free FFA exhibited a release rate of 91.34 ± 0.98% within 9 h [41]. Additionally, free FFA demonstrated a rapid release of 96.1 ± 1.20% within the first 7 h, which was significantly higher than that of FCSPA nanofibers, which released 88.14 ± 1.37% of the loaded FFA drug. Within the first 3 h, only 68.4% of the drug was released from the FCSPA nanofibers, a considerably lower rate than that of pure FFA. Moreover, the higher water-uptake capacity of the FCSPA nanofibers may have contributed to the accelerated release of FFA. Burst release, a phenomenon frequently observed in various drug-delivery systems, was evident. This effect can benefit specific applications, such as wound treatment and targeted drug delivery, where an initial high drug concentration is beneficial.

### 3.7. Antibacterial Activity

The antibacterial properties of FFA, CSPA-1, CSPA-2, CSPA-3, and FCSPA nanofibers against *S. aureus* (gram-positive) and *E. coli* (gram-negative) were evaluated using the colony-counting method (Figure 7a). Among them, FCSPA demonstrated a significantly higher inhibition rate than the blank CSPA-1, CSPA-2, and CSPA-3 nanofibers, highlighting the strong antibacterial performance of the FFA drug [42]. As shown in Figure 7b,c, the antibacterial efficacy of the FFA, CSPA-1, CSPA-2, CSPA-3, and FCSPA nanofibers ranged from 39.2% to 98.5% and 39.8% to 96.7% against *S. aureus* and *E. coli*, respectively. The inhibition rate of the FCSPA nanofibers reached 98.5 ± 3.89% against *S. aureus* and 96.7 ± 2.17% against *E. coli*, which were significantly higher than those of FFA (39.2 ± 1.75% and 39.8 ± 2.09%), CSPA-1 (54.3 ± 1.21% and 49.8 ± 2.39%), CSPA-2 (68.7 ± 2.18% and 63.6 ± 1.99%), and CSPA-3 (79.7 ± 2.06% and 76.5 ± 1.28%), respectively (Figure 7b,c) [10]. Interestingly, the antibacterial activity was more pronounced against *S. aureus* than against *E. coli*, which may be attributed to structural differences in the cell walls of gram-positive and gram-negative bacteria [23,33]. These findings suggest that the incorporation of FFA into CSPA-3 nanofibers significantly enhances their antimicrobial properties against both bacterial strains. Bacterial growth kinetics revealed that the standard antibiotics (vancomycin and ciprofloxacin) maintained the lowest OD_600_ values (0.05–0.1), indicating strong inhibition. FCSPA nanofibers showed moderate suppression with OD_600_ reaching 0.25 at 24 h, whereas free FFA exhibited the least effect with OD_600_ increasing to 0.89 (Figure 7d,e). These results highlight the enhanced and sustained antibacterial activity of the FCSPA nanofiber–based delivery system compared with that of free FFA.

### 3.8. Anticancer Analysis

Figure 8a shows the percentage of viable cells over 24 and 48 h of incubation. Compared with the control, the nanofibers demonstrated increased cell viability, with FFA obtaining 64.93 ± 2.14% and 68.69 ± 3.18%, CSPA-1 showing 61.37 ± 1.29% and 59.81 ± 2.47%, CSPA-2 at 56.31 ± 0.98% and 53.29 ± 2.07%, and CSPA-3 at 68.19 ± 2.62% and 71.25 ± 1.93% viability over the 24 and 48 h incubation periods, respectively [14]. Furthermore, the FCSPA nanofiber scaffolds exhibited significantly enhanced cell viability, reaching 73.02 ± 2.17% and 74.21 ± 1.67% at 24 and 48 h, respectively. However, a slight reduction in cell viability was observed in the FCSPA nanofibers, suggesting that FFA incorporation influenced the biocompatibility of the CSPA-3 nanofiber mats. This may be attributed to the decreased density of amino groups due to FFA grafting onto the CSPA backbone [37]. FFA appears to create a more conducive environment for cell attachment, differentiation, and proliferation owing to its inherent biocompatibility and biodegradability. These properties make it a promising candidate for biomedical applications, particularly in tissue engineering and regenerative medicine. Figure 8b illustrates the cellular behavior of HCT-116 cells cultured on the nanofiber scaffolds after 48 h of incubation. Cellular internalization was assessed using DAPI and PI staining, and the resulting images were captured using confocal laser scanning microscopy. The fluorescence images demonstrated that, compared with the control, the nanofiber scaffolds supported significantly higher cellular activity [43]. A greater number of HCT-116 cells adhered to and spread across the surfaces of FFA, CSPA-1, CSPA-2, CSPA-3, and FCSPA nanofibers, indicating enhanced cell–substrate interactions [44].

### 3.9. Antioxidant Analysis

The radical scavenging activity of newly fabricated materials is typically assessed using the DPPH radical entrapment method to evaluate their antioxidant properties in biological systems. The DPPH free radical scavenging activities of free FFA, CSPA-1, CSPA-2, CSPA-3, and FCSPA nanofibers exhibited time-dependent responses, with the scavenging rate increasing over time (6, 12, 18, and 24 h), indicating the sustained release of FFA from the CSPA (FCSPA) matrixes (Figure 9). Notably, the antioxidant activity of the FFA-loaded CSPA-3 nanofiber formulation (86.21 ± 2.36%) was significantly higher than that of the free FFA (48.5 ± 1.92%), CSPA-1 (39.8 ± 1.24%), CSPA-2 (61.7 ± 0.99%), and CSPA-3 (73.14 ± 3.17%) at 24 h. This enhancement can be attributed primarily to the antioxidant properties of CHS and the incorporation of FFA into the nanofiber structure [40,45]. The increased scavenging activity of FCSPA nanofibers underscores the significant contribution of FFA to its antioxidant performance. The notable antioxidant activity of the FFA-loaded CSPA-3 nanofibers was likely attributable to the high concentration of antioxidants present in the FFA. As the immersion time in the DPPH solution increased, more FFA was released from the nanofibers, resulting in a corresponding increase in the DPPH free radical scavenging activity [10,40,45]. The antioxidant activity of FFA is attributed to the presence of an –OH group at the carbon position. These findings indicate that the incorporation of FFA into the CSPA-3 nanofiber substantially enhances its antioxidant capacity, making it a promising candidate for biomedical applications requiring sustained antioxidant effects.

As presented in Table 2, the FFA-loaded CSPA nanofibers developed in this study demonstrated superior EE (91.2%) compared to other bioactive compound–loaded nanofiber systems reported previously [46,47,48,49]. Moreover, the sustained release of FFA over 72 h significantly exceeds the release durations typically observed for compounds like quercetin, ibuprofen, and resveratrol, which commonly exhibit burst or moderate-duration release profiles. The pronounced antioxidant activity and morphological integrity of the FCSPA nanofibers further highlight the effectiveness of the CSPA matrix in stabilizing and delivering FFA. These findings underscore the enhanced performance and potential applicability of the FFA nanofiber system in controlled drug delivery and bioactive wound-healing platforms.

## 4. Conclusions

In the present study, we encapsulated FFA in CSPA nanofibers via electrospinning, yielding FCSPA. The morphologies, chemical compositions, surface characteristics, and thermal properties of the prepared CSPA and FCSPA nanofibers were examined by SEM, FTIR, XRD, and DSC, respectively. The FESEM images of the CSPA and FCSPA nanofibers exhibited a narrow distribution, uniform structures, and a smooth morphology. The FCSPA nanofibers exhibited superior swelling capacity (302 ± 15.1% at 24 h), accelerated biodegradation (98.12 ± 3.31% at 72 h), and high EE% (91.2 ± 0.95%). The controlled drug-release experiments indicated the stability of FFA drugs that were released within 24 h (93.27 ± 2.31%). The FCSPA nanofibers inhibited *E. coli* (96.7 ± 2.17%) and *S. aureus* (98.5 ± 3.89%), and their antibacterial effects against gram-positive bacteria were better than those against gram-negative bacteria. Furthermore, their cell viabilities and anticancer activities indicated that the FCSPA nanofibers exhibited low toxicity and good anticancer activity against the HCT-116 cell line. The FCSPA nanofibers delivered the highest antioxidant performance among the tested samples, achieving an 86.21 ± 2.36% DPPH-scavenging activity in 24 h. These results highlight the fabricated FCSPA nanofiber as a promising candidate for wound-healing, drug-delivery, and tissue-engineering applications. However, because we have not comprehensively validated the aforementioned biomedical applications of our FCSPA nanofibers, we anticipate that further in vivo studies will focus on validating their clinical potential.

## Data Availability

The original contributions presented in this study are included in the article. Further inquiries can be directed to the corresponding authors.

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
