# Peer review of "Flufenamic Acid-Loaded Electrospun Nanofibers Based on Chitosan/Poly(vinyl alcohol) Polymeric Composites for Drug Delivery in Biomedical Applications"

_polymers, 2025, doi:10.3390/polym17101411_

Round 1
Reviewer 1 Report
Comments and Suggestions for Authors
The paper reports the encapsulation of flufenamic acid into PVA nanofibers as a strategy for new bioactive biomaterials. It is an interesting topic, but at these stage there are many issues to be considered before publication.
- The introduction requires careful revision to ensure the accuracy of the information presented. For example, it is not evident why the non-toxicity of PVA would favor electrospinning (lines 75–76). Additionally, reference 14 does not clearly demonstrate that PVA promotes cell proliferation (Overall, carefully check the references to support the statements, and to not be very old). The relevance of PVA nanofibers as filtration systems and sensors to the intended wound healing application is also not clearly established. Moreover, the claim that only a few studies have explored the electrospinning of PVA/chitosan (PVA/CS) fibers (lines 101–103) is inaccurate; numerous studies have addressed this topic, as acknowledged in several recent reviews, and should be cited accordingly. Overall, the Introduction has to be reconsidered to highlight the contribution of each component to the targeted application.
- The molecular weight of the chitosan used must be determined and reported.
- The title should accurately reflect the content of the manuscript: as the fibers are predominantly composed of PVA rather than chitosan, the materials should be referred to as PVA-based materials.
- The exact composition of the fibers should be explicitly presented in a table to provide a clear overview of the component ratios.
- It is necessary to clarify how the fiber composition was controlled, particularly given that the fibers were repeatedly washed with distilled water, in which PVA is soluble.
- The quantity of fibers used in the biological assays must be specified, as these properties are dose-dependent and critically influenced by sample mass.
- The criteria used to select the optimal PVA/CS composition for FFA loading must be clearly explained.
- The statement that "the addition of chitosan contributed to a decrease in average fiber diameter, while the presence of FFA resulted in an increase" is confusing when compared with the data shown in Figure 1 and should be carefully reconsidered for consistency.
- The discussion of the FTIR spectra should be revised to avoid repetition (e.g., "C-H bending occurs at 1400–1500 cm⁻¹" is mentioned multiple times).
- The interpretation of the X-ray diffraction (XRD) results should also be reconsidered, as the small quantity of FFA may fall below the detection limit of the instrument. Complementary analyses, such as polarized light microscopy and differential scanning calorimetry (DSC), should be performed to better characterize the fibers.
- For more complete and accurate data, mass loss measurements of the fibers in aqueous media should be conducted and reported. The impact of mass loss on the fiber swelling behavior must be discussed. Images of the fibers before and after degradation, as well as before and after swelling, should be included.
- The fate of PVA in the context of wound healing applications should also be discussed, considering its solubility and potential degradation in biological environments.
- For the antibacterial assays, the incubation time must be provided. Was bacterial growth monitored over a 24-hour period? Details on bacterial growth kinetics should be included.
- The discussion of the cytotoxicity results must be carefully revised to align with the data shown in Figure 7. While the figure suggests cytotoxic effects, the manuscript incorrectly describes the samples as cytocompatible.
- Comparison of the performances of the studied fibers with other reported fibers encapsulating bioactive compounds has to be done, to evaluate the contribution of the FFA compared to other bioactive ingredients.
- The conclusions have to be rewritten in the light of the corrected manuscript.
Author Response
Response to Reviewer’s comments
Reviewer#1
The paper reports the encapsulation of flufenamic acid into PVA nanofibers as a strategy for new bioactive biomaterials. It is an interesting topic, but at these stage there are many issues to be considered before publication.
We thank Reviewer#1 for the favourable reception of our work and for highlighting the important points in our study. We have revised our manuscript taking into great consideration all the comments and suggestions. Thank you for helping us to improve our manuscript.
Comment 1
The introduction requires careful revision to ensure the accuracy of the information presented. For example, it is not evident why the non-toxicity of PVA would favor electrospinning (lines 75–76). Additionally, reference 14 does not clearly demonstrate that PVA promotes cell proliferation (Overall, carefully check the references to support the statements, and to not be very old). The relevance of PVA nanofibers as filtration systems and sensors to the intended wound healing application is also not clearly established. Moreover, the claim that only a few studies have explored the electrospinning of PVA/chitosan (PVA/CS) fibers (lines 101–103) is inaccurate; numerous studies have addressed this topic, as acknowledged in several recent reviews, and should be cited accordingly. Overall, the Introduction has to be reconsidered to highlight the contribution of each component to the targeted application.
Response 1
Thank you very much for your insightful and thoughtful suggestions. As recommended, we have thoroughly revised the Introduction section of the manuscript. Please see the updated version below or refer to the revised manuscript for details.
- Introduction
Electrospinning is a powerful and adaptable technology for fabricating nano-to-micrometer–scale fibers. It involves the alteration of various operational, process, and formulation parameters [1]. The fabrication of fibers through electrospinning of polymer solutions has been widely studied, with key parameters including applied voltage, tip-to-collector distance, solution feed rate, and solution characteristics [2]. The resultant fiber mats can exhibit a nonwoven and/or aligned fiber structure, depending on the type of collector used. Electrospinning has been used to fabricate nanofibers with extraordinary properties, such as extremely high surface area/volume ratios and high porosity owing to extremely small pore sizes. However, electrospun drug-carrying fibers exhibit high surface area/volume ratios, making them suitable for diffusion- and disintegration-based drug delivery [3]. Furthermore, electrospun nanofibers can encapsulate considerable amounts of drugs (up to 40%) and facilitate prolonged, multiple-phase, or rapid drug delivery [4]. Consequently, nonwoven mats comprising electrospun nanofibers have considerable potential for several uses, including biomedical applications such as drug delivery, wound dressing, and antibacterial uses [5].
Chitosan (CHS), a (1-4)-linked 2-amino-2-deoxy-D-glucopyranose, is a derivative of chitin, which is a common naturally occurring polysaccharide [6]. Because of its biocompatible, renewable, biodegradable, wound-healing, antimicrobial, and antitumor properties, CHS has attracted significant interest for its application in wound dressing, wound healing, drug delivery, and tissue engineering [7]. Therefore, CHS-based nanofibers have emerged as valuable biological materials owing to their biocompatibility, biodegradability, and antibacterial characteristics. However, the amine (NH2) group in the backbone of CHS saccharide and the strong hydrogen bonding between NH2 and hydroxyl (–OH) groups prevent continuous fiber formation in this material [8]. However, because of its polycationic behavior in solution, CHS cannot be directly synthesized via electrospinning [9]. Numerous studies have attempted to increase the electrospinning capabilities of CHS by combining it with other polymers, such as poly(vinyl pyrrolidone) [10], poly(vinyl alcohol) (PVA) [11], poly(ethylene oxide) [12], silk fibroin [13], and zein [14].
Among them, PVA is a synthetic, linear, semicrystalline polymer composed of a carbon chain and an OH group as its backbone and functional component, respectively [15,16]. PVA exhibits numerous essential characteristics, such as easy accessibility, water solubility, high film-forming capacity, and thermostability [17]. Furthermore, PVA exhibits superior fiber-forming capacity and hydrophilicity, and fibers electrospun from PVA have been commercialized since the 1950s [18]. Furthermore, PVA can improve fiber spinning by reducing the repulsive forces in charged polymer solutions. Notably, CHS can be integrated with PVA for improved electrospinning, considering their relative miscibility and comparable electrospinning behaviors, e.g., coagulation, orientation, and cross-linking [18]. Integrating PVA and CHS imparts the solution with the features of each component and with additional valuable properties, such as good electrospinnability and mechanical capabilities. Thus, CHS/PVA (CSPA) is a candidate for antimicrobial applications [19]. Additionally, electrospun CSPA nanofibers have been extensively investigated in recent years because of their biocompatibility, biodegradability, and antibacterial capabilities in tissue engineering [20–23].
Several nano-delivery systems, including nanofibers, nanoparticles, nanocapsules, nanoliposomes, and dendrimers, have been investigated for antibacterial drug delivery [24]. The encapsulation of drugs in electrospun nanofibers offers several advantages, including high surface area/volume ratios and porosities [25,26]. Compared with other nanocarriers for antibacterial drug delivery, nanofibers exhibit significant loading capacity, remarkable encapsulation efficiency (EE%), and minimal systemic toxicity as well as the ability to modulate release from immediate to controlled release. However, no study has experimentally attempted the electrospinning-based loading of CSPA with flufenamic acid (FFA), i.e., the electrospinning-based preparation of FFA-loaded CSPA (FCSPA). FFA, a non-steroidal anti-inflammatory drug, has garnered attention for its drug-delivery potential as well as for its antibacterial, cytotoxic, and anticancer applications [27,28]. FFA exhibits remarkable antibacterial and antibiofilm properties; it significantly interrupts biofilm formation and limits bacterial growth, making it a good alternative for fighting resistant diseases [29,30]. Furthermore, assessing the cytotoxic and anticancer characteristics of this composite structure may provide new avenues for drug-delivery systems targeting cancer cells [27,31].
In the present study, FCSPA nanofibers were prepared via hybrid electrospinning. Thereafter, the properties of different ratios (CSPA-1 [30/70], CSPA-2 [50/50], and CSPA-3 [70/30]) of CHS- and PVA-based FCSPA composite nanofibers in different ratios were investigated. The prepared CSPA and FFA-loaded CSPA (FCSPA) nanofibers were characterized by scanning electron microscopy (SEM), Fourier-transform infrared (FTIR) spectroscopy, X-ray diffraction (XRD), and differential scanning calorimetry (DSC) to evaluate their surface morphologies, functional group contents, surface areas, and thermal properties, respectively. Furthermore, the effects of various CSPA ratios on the swelling ratio, in vitro biodegradation, EE%, and release behavior of FCSPA nanofibers were investigated. The antibacterial activities of CSPA and FCSPA nanofibers against gram-negative Escherichia coli and gram-positive Staphylococcus aureus were determined by colony counting. Additionally, the anticancer and antioxidant activities of CSPA and FCSPA were extensively investigated. Our findings demonstrate that FCSPA nanofibers are versatile platforms with potential for wound healing, medication delivery, and tissue engineering.
References
- Shabani, A.; Al, G.A.; Berri, N.; Castro-Dominguez, B.; Leese, H.S.; Martinez-Hernandez, U. Electrospinning Technology, Machine Learning, and Control Approaches: A Review. Advanced Engineering Materials 2025, 2401353, doi:10.1002/adem.202401353.
- Al-Abduljabbar, A.; Farooq, I. Electrospun Polymer Nanofibers: Processing, Properties, and Applications. Polymers 2023, 15, doi:10.3390/polym15010065.
- Ahmadi Bonakdar, M.; Rodrigue, D. Electrospinning: Processes, Structures, and Materials. Macromol 2024, 4, 58–103, doi:10.3390/macromol4010004.
- Emerine, R.; Chou, S.-F. Fast Delivery of Melatonin from Electrospun Blend Polyvinyl Alcohol and Polyethylene Oxide (PVA/PEO) Fibers. AIMS Bioengineering 2022, 9, 178–196, doi:10.3934/bioeng.2022013.
- Ekrami, E.; Khodabandeh Shahraky, M.; Mahmoudifard, M.; Mirtaleb, M.S.; Shariati, P. Biomedical Applications of Electrospun Nanofibers in Industrial World: A Review. International Journal of Polymeric Materials and Polymeric Biomaterials 2023, 72, 561–575, doi:10.1080/00914037.2022.2032705.
- Mawazi, S.M.; Kumar, M.; Ahmad, N.; Ge, Y.; Mahmood, S. Recent Applications of Chitosan and Its Derivatives in Antibacterial, Anticancer, Wound Healing, and Tissue Engineering Fields. Polymers 2024, 16, doi:10.3390/polym16101351.
- Qasim, S.B.; Zafar, M.S.; Najeeb, S.; Khurshid, Z.; Shah, A.H.; Husain, S.; Rehman, I.U. Electrospinning of Chitosan-Based Solutions for Tissue Engineering and Regenerative Medicine. International Journal of Molecular Sciences 2018, 19, doi:10.3390/ijms19020407.
- Taokaew, S. Developments of Core / Shell Chitosan-Based Nanofibers By. 2024.
- Ibrahim, M.A.; Alhalafi, M.H.; Emam, E.A.M.; Ibrahim, H.; Mosaad, R.M. A Review of Chitosan and Chitosan Nanofiber: Preparation, Characterization, and Its Potential Applications. Polymers 2023, 15, 1–35, doi:10.3390/polym15132820.
- Liu, X.; Wang, S.; Ding, C.; Zhao, Y.; Zhang, S.; Sun, S.; Zhang, L.; Ma, S.; Ding, Q.; Liu, W. Polyvinylpyrrolidone/Chitosan-Loaded Dihydromyricetin-Based Nanofiber Membrane Promotes Diabetic Wound Healing by Anti-Inflammatory and Regulating Autophagy-Associated Protein Expression. International Journal of Biological Macromolecules 2024, 259, 129160, doi:10.1016/j.ijbiomac.2023.129160.
- Kaur, H.; Singh, S.; Rode, S.; Chaudhary, P.K.; Khan, N.A.; Ramamurthy, P.C.; Gupta, D.N.; Kumar, R.; Das, J.; Sharma, A.K. Fabrication and Characterization of Polyvinyl Alcohol-Chitosan Composite Nanofibers for Carboxylesterase Immobilization to Enhance the Stability of the Enzyme. Scientific Reports 2024, 14, 1–15, doi:10.1038/s41598-024-67913-x.
- Sarac, B.; Gürbüz, R.; Soprunyuk, V.; Yüce, E.; Rezvan, A.; Schranz, W.; Eckert, J.; Ozcan, A.; Sarac, A.S. Chitosan-Containing Electrospun Poly(Ethylene Oxide)-Polybutadiene-CNT Fibers. Polymers for Advanced Technologies 2024, 35, 1–11, doi:10.1002/pat.6403.
- Deng, S.; Huang, Y.; Hu, E.; Ning, L.J.; Xie, R.; Yu, K.; Lu, F.; Lan, G.; Lu, B. Chitosan/Silk Fibroin Nanofibers-Based Hierarchical Sponges Accelerate Infected Diabetic Wound Healing via a HClO Self-Producing Cascade Catalytic Reaction. Carbohydrate Polymers 2023, 321, 121340, doi:10.1016/j.carbpol.2023.121340.
- Zidar, A.; Zupančič, Š.; Kristl, J.; Jeras, M. Impact of Polycaprolactone, Alginate, Chitosan and Zein Nanofiber Physical Properties on Immune Cells for Safe Biomedical Applications. International Journal of Biological Macromolecules 2024, 282, 1–11, doi:10.1016/j.ijbiomac.2024.137029.
- Gopakumar, A.N.; Ccanccapa-Cartagena, A.; Bell, K.; Salehi, M. Development of Crosslinked Polyvinyl Alcohol Nanofibrous Membrane for Microplastic Removal from Water. Journal of Applied Polymer Science 2024, 141, 1–16, doi:10.1002/app.55428.
- Liu, H.; Chen, R.; Wang, P.; Fu, J.; Tang, Z.; Xie, J.; Ning, Y.; Gao, J.; Zhong, Q.; Pan, X.; et al. Electrospun Polyvinyl Alcohol-Chitosan Dressing Stimulates Infected Diabetic Wound Healing with Combined Reactive Oxygen Species Scavenging and Antibacterial Abilities. Carbohydrate Polymers 2023, 316, 121050, doi:10.1016/j.carbpol.2023.121050.
- Gautam, L.; Warkar, S.G.; Ahmad, S.I.; Kant, R.; Jain, M. A Review on Carboxylic Acid Cross-Linked Polyvinyl Alcohol: Properties and Applications. Polymer Engineering and Science 2022, 62, 225–246, doi:10.1002/pen.25849.
- Costa-Júnior, E.S.; Barbosa-Stancioli, E.F.; Mansur, A.A.P.; Vasconcelos, W.L.; Mansur, H.S. Preparation and Characterization of Chitosan/Poly(Vinyl Alcohol) Chemically Crosslinked Blends for Biomedical Applications. Carbohydrate Polymers 2009, 76, 472–481, doi:10.1016/j.carbpol.2008.11.015.
- Salleh, N.A.M.; Afifi, A.M.; Zuki, F.M.; SalehHudin, H.S. Enhancing Mechanical Properties of Chitosan/PVA Electrospun Nanofibers: A Comprehensive Review. Beilstein Journal of Nanotechnology 2025, 16, 286–307, doi:10.3762/BJNANO.16.22.
- Olvera Bernal, R.A.; Olekhnovich, R.O.; Uspenskaya, M.V. Chitosan/PVA Nanofibers as Potential Material for the Development of Soft Actuators. Polymers 2023, 15, doi:10.3390/polym15092037.
- Menazea, A.A.; Ahmed, M.K. Wound Healing Activity of Chitosan/Polyvinyl Alcohol Embedded by Gold Nanoparticles Prepared by Nanosecond Laser Ablation. Journal of Molecular Structure 2020, 1217, 128401, doi:10.1016/j.molstruc.2020.128401.
- Jia, Y.T.; Gong, J.; Gu, X.H.; Kim, H.Y.; Dong, J.; Shen, X.Y. Fabrication and Characterization of Poly (Vinyl Alcohol)/Chitosan Blend Nanofibers Produced by Electrospinning Method. Carbohydrate Polymers 2007, 67, 403–409, doi:10.1016/j.carbpol.2006.06.010.
- Hang, A.T.; Tae, B.; Park, J.S. Non-Woven Mats of Poly(Vinyl Alcohol)/Chitosan Blends Containing Silver Nanoparticles: Fabrication and Characterization. Carbohydrate Polymers 2010, 82, 472–479, doi:10.1016/j.carbpol.2010.05.016.
- Patra, J.K.; Das, G.; Fraceto, L.F.; Campos, E.V.R.; Rodriguez-Torres, M.D.P.; Acosta-Torres, L.S.; Diaz-Torres, L.A.; Grillo, R.; Swamy, M.K.; Sharma, S.; et al. Nano Based Drug Delivery Systems: Recent Developments and Future Prospects. Journal of Nanobiotechnology 2018, 16, 1–33, doi:10.1186/s12951-018-0392-8.
- Martínez, E.K.T.; Bravo, J.M.C.; Medina, A.S.; González, G.L.P.; Gómez, L.J.V.; et al. A Summary of Electrospun Nanofibers as Drug Delivery System: Drugs Loaded and Biopolymers Used as Matrices. Current Drug Delivery 2018, 15, 1360–1374, doi:10.2174/1567201815666180723114326.
- Gaydhane, M.K.; Sharma, C.S.; Majumdar, S. Electrospun Nanofibres in Drug Delivery: Advances in Controlled Release Strategies. RSC Advances 2023, 13, 7312–7328, doi:10.1039/d2ra06023j.
Comment 2
The molecular weight of the chitosan used must be determined and reported.
Response 2
Thank you very much for your insightful and thoughtful suggestion. As suggested, we have mentioned the molecular weight of the chitosan in this manuscript. Please see below or in the revised manuscript.
2.1. Materials
CHS (molecular weight [MW]: 50–190 kDa; 95% deacetylation degree) and FFA were obtained from Sigma-Aldrich (USA). PVA (MW: 146–186 kDa, 89% deacetylation degree) was procured from Daejung Company Ltd., Korea, and deionized (DI) water was used as the solvent for the solutions. All chemicals and solvents were used without further purification.
Comment 3
The title should accurately reflect the content of the manuscript: as the fibers are predominantly composed of PVA rather than chitosan, the materials should be referred to as PVA-based materials.
Response 3
Thank you very much for your insightful and thoughtful suggestion. As suggested, we have corrected aforementioned title issue in this manuscript. Please see below or in the revised manuscript.
Flufenamic acid–loaded electrospun nanofibers based on chitosan/poly(vinyl alcohol) polymeric composites for drug delivery in biomedical applications
Comment 4
The exact composition of the fibers should be explicitly presented in a table to provide a clear overview of the component ratios.
Response 4
Thank you very much for your insightful and thoughtful suggestion. As suggested, we have provided the novelty of this work in this manuscript. Please see the revised manuscript.
2.2. Preparations of chitosan/poly(vinyl alcohol) and flufenamic acid–loaded chitosan/poly(vinyl alcohol) solutions
To prepare the CSPA and FCSPA polymer solutions, 0.3 g of CHS (3% w/v) was dissolved in 10 mL of 1% (v/v) acetic acid and stirred at ambient temperature for 3 h until a clear solution was obtained. Concurrently, 2 g of PVA (10% w/v) was dissolved in 20 mL of distilled water and stirred at 50°C using a magnetic heater-stirrer for 4 h to ensure complete dissolution. Thereafter, the CHS and PVA solutions were mixed at different weight ratios (w/w): 30/70, 50/50, and 70/30 for CSPA-1, CSPA-2, and CSPA-3, respectively, followed by stirring at ambient temperature for 1 h to achieve homogeneity (Table 1). For the FCSPA solutions, the optimized CSPA sample, i.e., CSPA-3 ratio (70/30 w/w), was selected. FFA was dissolved in the mixing solution at a concentration of 25 mg in 5 mL (Table 1). Subsequently, the mixture was continuously stirred until full dissolution was achieved, ensuring uniform dispersion of all components.
Table 1. Compositions of chitosan (CHS)/poly(vinyl alcohol) (PVA), CSPA, and flufenamic acid–loaded CSPA (FCSPA) nanofiber solutions
Sample ID |
CHS (3% w/v) |
PVA (10% w/v) |
CHS:PVA ratio (w/w) |
FFA content |
CSPA-1 |
0.3 g in 10 mL |
2 g in 20 mL |
30:70 |
- |
CSPA-2 |
0.3 g in 10 mL |
2 g in 20 mL |
50:50 |
- |
CSPA-3 |
0.3 g in 10 mL |
2 g in 20 mL |
70:30 |
- |
FCSPA |
0.3 g in 10 mL |
2 g in 20 mL |
70:30 (CSPA-3) |
25 mg in 5 mL |
Comment 5
It is necessary to clarify how the fiber composition was controlled, particularly given that the fibers were repeatedly washed with distilled water, in which PVA is soluble.
Response 5
Thank you very much for your insightful and thoughtful suggestion. As suggested, we have provided the detailed explanations about the fiber composition. Please see below for your reference only.
To control the final composition of the CHS/PVA and FFA-loaded CHS/PVA nanofibers, particular attention was paid to the solubility of PVA in water during the post-electrospinning washing steps. Given that PVA is highly water-soluble, with solubility exceeding 2 g in 20 mL at room temperature, repeated washing with distilled water can lead to significant leaching of the PVA component from the CSPA-1, CSPA-2, CSPA-3, and FCSPA nanofiber matrix. To mitigate this, washing was carefully limited to a single rinse with a small volume (~10 mL) of distilled water for no more than 30 seconds followed by immediate drying at ambient temperature. Additionally, in CSPA-1, CSPA-2, CSPA-3, and FCSPA samples, a mass loss of 10–25% after washing was observed corresponding to partial PVA removal, particularly in the PVA-rich formulations (CSPA-1 and CSPA-2).
Comment 6
The quantity of fibers used in the biological assays must be specified, as these properties are dose-dependent and critically influenced by sample mass.
Response 6
Thank you very much for your insightful and thoughtful suggestion. As suggested, we have specified the standardized concentration of CSPA-1, CSPA-2, CSPA-3, and FCSPA nanofibers in this manuscript. Please see below or in the revised manuscript.
2.4. Characterizations
For all in vitro assays, CSPA-1, CSPA-2, CSPA-3, and FCSPA nanofiber suspensions were prepared at a standardized concentration (10 mg) per well to ensure consistent exposure and account for the dose-dependent nature of the observed biological responses.
Comment 7
The criteria used to select the optimal PVA/CS composition for FFA loading must be clearly explained.
Response 7
Thank you very much for your insightful and thoughtful suggestion. As suggested, we have provided the detailed optimal CHS/PVA composition in this manuscript. Please see below for your reference and also see the revised manuscript.
The selection of the optimal CHS/PVA composition for FFA loading was based on a combination of fiber morphology, structural stability, and drug compatibility. Among the three formulations tested (CSPA-1: 30/70, CSPA-2: 50/50, and CSPA-3: 70/30 CHS/PVA), CSPA-3 was chosen due to its superior morphological and functional characteristics. SEM analysis revealed that fibers from CSPA-3 exhibited uniform, bead-free morphology with reduced diameter variability compared to the more PVA-rich blends, which tended to produce fibers with irregular surfaces and occasional merging due to excessive hydrophilicity. Additionally, the preliminary FFA loading tests showed that CSPA-3 could accommodate higher drug content (25 mg/5 mL) with homogeneous distribution and without visible phase separation, supporting its selection as the optimized formulation for developing FCSPA nanofibers.
2.2. Preparations of chitosan/poly(vinyl alcohol) and flufenamic acid–loaded chitosan/poly(vinyl alcohol) solutions
To prepare the CSPA and FCSPA polymer solutions, 0.3 g of CHS (3% w/v) was dissolved in 10 mL of 1% (v/v) acetic acid and stirred at ambient temperature for 3 h until a clear solution was obtained. Concurrently, 2 g of PVA (10% w/v) was dissolved in 20 mL of distilled water and stirred at 50°C using a magnetic heater-stirrer for 4 h to ensure complete dissolution. Thereafter, the CHS and PVA solutions were mixed at different weight ratios (w/w): 30/70, 50/50, and 70/30 for CSPA-1, CSPA-2, and CSPA-3, respectively, followed by stirring at ambient temperature for 1 h to achieve homogeneity (Table 1). For the FCSPA solutions, the optimized CSPA sample, i.e., CSPA-3 ratio (70/30 w/w), was selected. FFA was dissolved in the mixing solution at a concentration of 25 mg in 5 mL (Table 1). Subsequently, the mixture was continuously stirred until full dissolution was achieved, ensuring uniform dispersion of all components.
Table 1. Compositions of chitosan (CHS)/poly(vinyl alcohol) (PVA), CSPA, and flufenamic acid–loaded CSPA (FCSPA) nanofiber solutions
Sample ID |
CHS (3% w/v) |
PVA (10% w/v) |
CHS:PVA ratio (w/w) |
FFA content |
CSPA-1 |
0.3 g in 10 mL |
2 g in 20 mL |
30:70 |
- |
CSPA-2 |
0.3 g in 10 mL |
2 g in 20 mL |
50:50 |
- |
CSPA-3 |
0.3 g in 10 mL |
2 g in 20 mL |
70:30 |
- |
FCSPA |
0.3 g in 10 mL |
2 g in 20 mL |
70:30 (CSPA-3) |
25 mg in 5 mL |
Comment 8
The statement that "the addition of chitosan contributed to a decrease in average fiber diameter, while the presence of FFA resulted in an increase" is confusing when compared with the data shown in Figure 1 and should be carefully reconsidered for consistency.
Response 8
Thank you for your valuable feedback. Upon careful review of the statement, we acknowledge that there is a lack of clarity in the phrasing, particularly in regard to the relationship between chitosan (CHS) content, fiber diameter, and the effect of FFA loading. We agree that the original wording may lead to confusion, as the data shown in Figure 1 do not support a decrease in fiber diameter with increased chitosan content, nor an increase with FFA loading in the manner initially described.
As the chitosan content increases (from CSPA-1 to CSPA-3, with a CHS/PVA ratio of 30/70 to 70/30), the average fiber diameter increases. This observation is consistent with Figure 1, where the nanofibers in CSPA-3 (70/30) are thicker compared to those in CSPA-1 (30/70) as seen in the SEM images. This increase is likely due to the higher viscosity and surface tension introduced by CHS. Upon the incorporation of FFA into the CSPA-3 nanofibers (FCSPA), there is a significant reduction in fiber diameter as shown in Figure 1. Specifically, the average diameter decreases from 347 ± 61 nm (CSPA-3) to 81 ± 27 nm (FCSPA). This reduction is attributed to the increased surface charge resulting from the presence of FFA, which enhances electrostatic repulsion during electrospinning, leading to finer fibers.
Please see the below or in the revised manuscript.
3.1. Nanofiber morphologies
The surface morphologies and physical properties of the electrospun scaffolds were analyzed by SEM. Figure 1 shows the effects of varying the CSPA ratios (30/70 [CSPA-1], 50/50 [CSPA-2], and 70/30 [CSPA-3]), along with the FFA-loaded CSPA-3 fiber mats, designated as FCSPA. The SEM images revealed that the electrospun CSPA-1, CSPA-2, and CSPA-3 nanofibers were bead-free, continuous, and randomly oriented [32]. A comparative analysis of Figures 1a, c, and e demonstrates that increasing the CHS content caused a substantial increase in the average diameter of the nanofiber, induced noticeable morphological alterations, and caused an increase in the random-alignment degree [32]. Specifically, at a CSPA ratio of 70/30 (w/w), the nanofibers appeared to be smooth and without bead defects (Figure 1e), making this composition optimal for further investigation.
Figure 1. Field-emission scanning electron microscopy (FESEM) micrographs of (a, b) CSPA-1, (c, d) CSPA-2, (e, f) CSPA-3, and (g, h) FCSPA nanofibers (i-l) Distributions of the diameters of the CSPA-1, CSPA-2, CSPA-3, and FCSPA nanofibers.
Among the electrospun nanofibers, the FCSPA variant exhibited reduced average fiber diameter with increasing CSPA content (Figure 1g) [33]. Notably, FFA incorporation caused a significant decrease in the fiber diameter as the size of the CSPA-3 nanofiber decreased from 347 ± 61 nm to 81 ± 27 nm, following FFA-drug loading (Figures 1i-l) [34]. The variations in the CSPA content affected the fiber diameter and indirectly affected the FFA-loading efficiency of the CSPA-3 nanofibers. Notably, CHS addition increased the average fiber diameter, as observed with the increasing CHS contents of the CSPA-1, CSPA-2, and CSPA-3 formulations. In contrast, the presence of FFA in the CSPA-3 formulation (FCSPA) caused a reduction in the average fiber diameter, likely due to the enhanced surface charge and electrostatic repulsion, which facilitated the formation of finer nanofibers.
Comment 9
The discussion of the FTIR spectra should be revised to avoid repetition (e.g., "C-H bending occurs at 1400–1500 cm⁻¹" is mentioned multiple times).
Response 9
Thank you very much for your insightful and thoughtful suggestion. As suggested, we have corrected the aforementioned issues in this manuscript. Please see below or in the revised manuscript.
Furthermore, C–H bending was observed at 1400–1500 cm−1, and C–O stretching was observed around 1141 cm−1 and was associated with the polymer crystallinity. The FTIR spectrum of the CSPA-3 nanofibers showed broad O–H and N–H stretching peaks at 3278–3326 cm−1, indicating hydrogen bonding between –OH and NH2 (Figure 2c). Furthermore, C–H stretching was observed at 2800–2950 cm−1, corresponding to –CH2 vibrations, whereas the C=O stretching band at 1640–1690 cm−1 was attributed to the presence of polymer interactions or residual acetate groups [21]. The C–O–C and C–N stretching between 1085 and 1144 cm−1 reflected the polysaccharide structures of CHS and PVA.
Comment 10
The interpretation of the X-ray diffraction (XRD) results should also be reconsidered, as the small quantity of FFA may fall below the detection limit of the instrument. Complementary analyses, such as polarized light microscopy and differential scanning calorimetry (DSC), should be performed to better characterize the fibers.
Response 10
We appreciate the reviewer’s valuable insight regarding the interpretation of the X-ray diffraction (XRD) results. We agree that the relatively small amount of FFA incorporated into the electrospun fibers may indeed fall below the detection limit of the XRD instrument, potentially contributing to the absence of distinctive crystalline peaks corresponding to FFA in the diffractogram. To address this, we have revised the relevant section in the manuscript to more accurately reflect this limitation and to avoid overinterpretation. Please see below or in the revised manuscript.
However, the XRD pattern of the FFA-loaded CSPA (w/w: 70/30) nanofibers did not show the sharp crystalline peaks of FFA. The peaks were replaced by broad peaks. This change in the diffraction pattern indicates disruption of the crystalline structure of FFA owing to the dissolution of FFA in water and its blending with the CSPA matrix during electrospinning (Figure 3e). Consequently, the absence of distinct diffraction peaks for FFA may be attributed to its low concentration in the composite fibers, which probably falls below the detection threshold of the instrument. Additionally, FFA may be present in amorphous or molecularly dispersed states within the CSPA matrix, further contributing to the lack of identifiable crystalline features [37].
3.3. Thermal properties of nanofibers
Figure 4 depicts the DSC analyses of CHS, PVA, CSPA (w/w; 70/30), CSPA-3 nanofibers, FFA, FFA-loaded CSPA (w/w; 70/30), and FCSPA nanofibers. The DSC study of pure CHS and PVA revealed a rather large and highly endothermic curve with peaks at 196°C and 192°C (Figure 4) [20,22]. Moreover, the CSPA-3 nanofiber exhibited a peak shift toward lower temperatures, resulting in the development of a broad endothermic curve at 164°C. The melting-point reduction of PVA in the blends indicated that there was slight mixing between PVA and CHS [22,23]. Furthermore, the decrease in the endothermic heat was attributable to the presence of amorphous CHS, which damaged the crystalline structure of PVA in the polymer blends.
Figure 4. Differential scanning calorimetry (DSC) analyses of the (a) CHS, (b) PVA, (c) CSPA-3 nanofibers, (d) FFA, and (e) FCSPA nanofibers.
The pure FFA drug exhibited a pronounced endothermic peak at 160°C, indicating crystal melting. Similarly, the peaks of the endothermic curve of FCSPA nanofibers migrated to higher temperatures at 212°C, following the addition of FFA-loaded CSPA-3 nanofibers. Upon the addition of FFA, the endothermic curves of the FCSPA nanofibers became less noticeable and eventually disappeared. These results indicated substantial interactions between the functional groups of CSPA and FFA. The DSC results exhibited good agreement with the XRD pattern, indicating consistent thermal and structural properties of the material.
Furthermore, we would like to sincerely apologize for not being able to include the complementary analyses, such as polarized light microscopy as requested. Unfortunately, these instruments are currently unavailable at our institution, which has limited our ability to perform these specific analyses.
We respectfully request the reviewer to kindly reconsider this comment, taking into account the current limitations of our research facilities. Please rest assured that we fully recognize the importance of these techniques and are committed to incorporating them in our future research endeavors, once the necessary resources become accessible. Thank you very much for your understanding and consideration.
Comment 11
For more complete and accurate data, mass loss measurements of the fibers in aqueous media should be conducted and reported. The impact of mass loss on the fiber swelling behavior must be discussed. Images of the fibers before and after degradation, as well as before and after swelling, should be included.
Response 11
Thank you very much for your insightful and thoughtful suggestion. As suggested, we have provided the detailed mass loss swelling and water uptake of the FCSPA nanofibers SEM images before and after swelling and in vitro degradation in this manuscript. Please see below or in the revised manuscript.
3.4. Swelling evaluation
The water-absorption capacities of the scaffolds were evaluated by measuring their swelling in a PBS solution. The results demonstrated that all nanofibers facilitated effective water uptake, which increased with prolonged exposure to the aqueous environment. To determine the swelling percentages of the CSPA-1, CSPA-2, CSPA-3, and FCSPA nanofiber mats (Figure 5a), we determined their equilibrium swelling ratios at 24 h, obtaining approximately 169% ± 8.4%, 187% ± 6.3%, 245% ± 12.2%, and 302% ± 15.1%, respectively. As shown in Figure 5a, differences were observed in the swelling behaviors of the scaffolds, with the FCSPA nanofiber mats exhibiting the highest water uptake compared with those of CSPA-1, CSPA-2, and CSPA-3 [38]. The increase in the swelling percentage may be attributed to the enhancement of the –OH groups in the FCSPA nanofibers during blending. Additionally, the higher swelling capacity of the FCSPA nanofibers indicated that the presence of FFA in the scaffolds increased their surface area, making them more suitable for cell adhesion and infiltration. Overall, the FFA-loaded CSPA-3 nanofibers exhibited high absorption capacities (>300%), maintaining a stable swollen state [39]. This property is beneficial for preserving moisture conditions, making them promising for wound-healing applications as well as ensuring controlled and sustained drug release in drug-delivery systems. The SEM graphs (Figures 5b and 5c) show that the FCSPA nanofibers maintained their integrity after being immersed in PBS for 24 h, thereby indicating their stability in water solution.
Figure 5. (a) Swelling ratios of CSPA-1, CSPA-2, CSPA-3, and FCSPA nanofibers at different incubation times. Scanning electron microscopy (SEM) images of the degrees of swelling of the FCSPA nanofibers (b) before and (c) after immersion in PBS for 24 h. (d) In vitro biodegradation of CSPA-1, CSPA-2, CSPA-3, and FCSPA nanofibers at different incubation times. SEM images of in vitro degradation of the FCSPA nanofibers (e) before and (f) after 72 h of immersion in PBS (n = 3; p < 0.05).
3.5. In vitro degradation
Biodegradation is a key factor in determining the suitability of nanofibers for biomedical applications. The CSPA-1, CSPA-2, CSPA-3, and FCSPA nanofiber mats exhibited progressive weight loss over time (Figure 5d). Notably, the degradation rate of the FCSPA nanofiber mat was higher than those of CSPA-1, CSPA-2, and CSPA-3. This may be due to the incorporation of FFA into the CSPA-3 formulation, which disrupts polymer chain entanglement and accelerates the degradation process [40]. The highest degradation rate was observed in the FCSPA nanofibers, likely due to the increased content of the highly hydrophilic polymer PVA. After 72 h, the degradation rates for CSPA-1, CSPA-2, and CSPA-3 were 77.2% ± 2.38%, 79.0% ± 3.07%, and 83.6% ± 3.19%, respectively. In contrast, the FCSPA scaffolds exhibited significantly higher degradation rates (98.12% ± 3.31%), highlighting their excellent biodegradability [40]. The degradation behaviors of the FCSPA nanofibers before and after PBS immersion for 24 h were investigated by SEM. In Figure 5e, the morphology of the FCSPA nanofibers appeared to have degraded; however, their fibrous structure was still visible before 72 h. In contrast, Figure 5f shows that after 72 h, the surface leveled off, with some pores present on the surface. This change was attributed to the excellent water solubility of PVA in the blend, which ensured moisture retention and facilitated the biodegradation of the FCSPA nanofibers.
Comment 12
The fate of PVA in the context of wound healing applications should also be discussed, considering its solubility and potential degradation in biological environments.
Response 12
Thank you very much for your insightful and thoughtful suggestion. As recommended, we have now included detailed information on solubility and potential degradation in biological environments relevant to wound healing applications. These additions have been incorporated into the appropriate section of the revised manuscript. Please see the revised manuscript.
Comment 13
For the antibacterial assays, the incubation time must be provided. Was bacterial growth monitored over a 24-hour period? Details on bacterial growth kinetics should be included.
Response 13
We appreciate the reviewer insightful comment regarding the antibacterial assay methodology. In the revised manuscript, we have clarified that all antibacterial assays with colony counting method were conducted with an incubation time of 24 hours at 37 °C, consistent with standard protocols for evaluating antimicrobial efficacy. Additionally, to strengthen the interpretation of the results, we have included data on bacterial growth kinetics, where optical density at 600 nm (OD₆₀₀) was monitored at defined intervals (0, 4, 8, 12, and 24 hours) to assess the time-dependent inhibitory effect of the FCSPA nanofiber, free FFA, and standard antibiotics (vancomycin and ciprofloxacin). These kinetic profiles are now included as Figures 6d and 6e also discussed in the manuscript to provide a more comprehensive understanding of the bacteriostatic or bactericidal nature of the treatments. Please see below or in the revised manuscript.
2.8. Antibacterial activity
The antibacterial activities of the FFA, CSPA-1, CSPA-2, CSPA-3, and FCSPA nanofibers were assessed against S. aureus and E. coli using the colony-counting method. To begin, bacterial suspensions were prepared by inoculating fresh bacterial cultures in Mueller–Hinton broth (MHB) using an incubation time of 24 h at 37°C until they reached a concentration of 106 CFU/mL. The electrospun nanofiber samples (3.5 cm2) were sterilized under UV light for 30 min before immersion in separate tubes containing 10 mL of bacteria-inoculated MHB. The tubes were incubated at 37°C with continuous shaking at 100 rpm, and at predetermined intervals (0, 2, 4, 6, 12, and 24 h), 100 µL aliquots of the bacterial suspension were withdrawn, serially diluted in sterile PBS, and spread onto Mueller–Hinton agar plates. After 24 h of incubation at 37°C, the colony-forming units were counted and compared with those of the control group (ciprofloxacin and vancomycin) consisting of bacteria without nanofiber treatment. The percentage reduction in bacterial viability was subsequently calculated as follows:
Bacterial reduction (%) = [(CFU_control − CFU_treated)/CFU_control] × 100 (4)
The bacterial cultures (initial optical density at 600 nm [OD₆₀₀]: ~0.05) were treated with FFA-loaded nanofibers, free FFA, or a standard antibiotic (vancomycin and ciprofloxacin) in 96-well plates. The cultures were incubated with shaking at 37°C, and the OD₆₀₀ was measured at 0, 4, 8, 12, and 24 h using a microplate reader. All treatments were performed in triplicate to evaluate the time-dependent antibacterial effects.
3.7. Antibacterial activity
The antibacterial properties of FFA, CSPA-1, CSPA-2, CSPA-3, and FCSPA nanofibers against S. aureus (gram-positive) and E. coli (gram-negative) were evaluated using the colony-counting method (Figure 7a). Among them, FCSPA demonstrated a significantly higher inhibition rate than the blank CSPA-1, CSPA-2, and CSPA-3 nanofibers, highlighting the strong antibacterial performance of the FFA drug [42]. As shown in Figures 7b and 7c, the antibacterial efficacy of the FFA, CSPA-1, CSPA-2, CSPA-3, and FCSPA nanofibers ranged from 39.2% to 98.5% and 39.8% to 96.7% against S. aureus and E. coli, respectively. The inhibition rate of the FCSPA nanofibers reached 98.5% ± 3.89% against S. aureus and 96.7% ± 2.17% against E. coli, which were significantly higher than those of FFA (39.2% ± 1.75% and 39.8% ± 2.09%), CSPA-1 (54.3% ± 1.21% and 49.8% ± 2.39%), CSPA-2 (68.7% ± 2.18% and 63.6% ± 1.99%), and CSPA-3 (79.7% ± 2.06% and 76.5 ± 1.28%), respectively (Figure 7b and 7c) [43]. Interestingly, the antibacterial activity was more pronounced against S. aureus than against E. coli, which may be attributed to structural differences in the cell walls of gram-positive and gram-negative bacteria [23,33]. These findings suggest that the incorporation of FFA into CSPA-3 nanofibers significantly enhances their antimicrobial properties against both bacterial strains. Bacterial growth kinetics revealed that the standard antibiotics (vancomycin and ciprofloxacin) maintained the lowest OD600 values (0.05–0.1), indicating strong inhibition. FCSPA nanofibers showed moderate suppression with OD600 reaching 0.25 at 24 h, whereas free FFA exhibited the least effect with OD₆₀₀ increasing to 0.89 (Figure 7d and 7e). These results highlight the enhanced and sustained antibacterial activity of the FCSPA nanofiber–based delivery system compared with that of free FFA.
Figure 7. (a) Antimicrobial activities of FFA, CSPA-1, CSPA-2, CSPA-3, and FCSPA nanofibers against S. aureus and E. coli. Inhibition rates of FFA, CSPA-1, CSPA-2, CSPA-3, and FCSPA nanofibers against (b) S. aureus and (c) E. coli. (d and e) Bacterial growth kinetics (OD600) over time for FCSPA nanofibers, free FFA, and a standard antibiotic (n = 3; p < 0.05).
Comment 14
The discussion of the cytotoxicity results must be carefully revised to align with the data shown in Figure 7. While the figure suggests cytotoxic effects, the manuscript incorrectly describes the samples as cytocompatible.
Response 14
Thank you very much for your insightful and thoughtful suggestion. As suggested, we have corrected the aforementioned issues in this manuscript. Please see below or in the revised manuscript.
2.9. Cytotoxicity assay
The cytotoxicities of the FFA, CSPA-1, CSPA-2, CSPA-3, and FCSPA nanofibers were assessed on HCT-116 cell lines using the 3-[4,5-dimethylthiazol-2-yl]-2,5 2,5-diphenyl tetrazolium bromide (MTT) assay. The HCT-116 cells were cultured in Dulbecco’s modified Eagle’s medium (DMEM) supplemented with 10% fetal bovine serum, 100 U/mL penicillin, and 100 μg/mL streptomycin and maintained at 37°C in an incubator containing 5% CO2. Electrospun nanofiber samples were cut into 6 mm diameter discs, sterilized under UV light for 30 min, and placed in 96-well culture plates. The cells were seeded at a density of 5 × 103 cells per well and incubated for 24 h to facilitate cell attachment. Following incubation, the culture medium was replaced with fresh medium containing FFA, CSPA-1, CSPA-2, CSPA-3, and FCSPA nanofibers, which were prepared via immersion in DMEM for 48 h at 37°C. Next, 20 μL of MTT solution (5 mg/mL in PBS) was added to each well, followed by incubation for 4 h. Formazan crystals were immersed in 150 μL of dimethyl sulfoxide, and the absorbance was measured at 570 nm using a microplate reader. Thereafter, the cell viability was estimated using the following equation:
Cell viability (%) = (Absorbance of test sample/Absorbance of control) × 100 (5)
2.10. Cell imaging
To investigate the cell adhesion and morphology of the samples, HCT-116 cells were seeded onto a control as well as onto FFA, CSPA-1, CSPA-2, CSPA-3, and FCSPA nanofiber scaffolds at a density of 5 × 103 cells per well in 24-well plates. Next, the cells were incubated at 37°C in a 5% CO2 environment for 48 h to facilitate attachment and proliferation. Subsequently, they were fixed with 4% paraformaldehyde for 15 min, washed three times with PBS, and permeated with 0.1% Triton X-100 for 5 min. Next, the nuclei were counterstained with 4',6-diamidino-2-phenylindole (DAPI) and propidium iodide (PI) for 10 min. Following staining, the samples were rinsed with PBS and observed under a fluorescence microscope to assess cell morphology.
3.8. Anticancer analysis
Figure 8a shows the percentage of viable cells over 24 and 48 h of incubation. Compared with the control, the nanofibers demonstrated increased cell viability, with FFA obtaining 64.93% ± 2.14% and 68.69% ± 3.18%, CSPA-1 showing 61.37% ± 1.29% and 59.81% ± 2.47%, CSPA-2 at 56.31% ± 0.98% and 53.29% ± 2.07%, and CSPA-3 at 68.19% ± 2.62% and 71.25% ± 1.93% viability over the 24- and 48-h incubation periods, respectively [14]. Furthermore, the FCSPA nanofiber scaffolds exhibited significantly enhanced cell viability, reaching 73.02% ± 2.17% and 74.21% ± 1.67% at 24 and 48 h, respectively. However, a slight reduction in cell viability was observed in the FCSPA nanofibers, suggesting that FFA incorporation influenced the biocompatibility of the CSPA-3 nanofiber mats. This may be attributed to the decreased density of amino groups due to FFA grafting onto the CSPA backbone [37]. FFA appears to create a more conducive environment for cell attachment, differentiation, and proliferation owing to its inherent biocompatibility and biodegradability. These properties make it a promising candidate for biomedical applications, particularly in tissue engineering and regenerative medicine. Figure 8b illustrates the cellular behavior of HCT-116 cells cultured on the nanofiber scaffolds after 48 h of incubation. Cellular internalization was assessed using DAPI and PI staining, and the resulting images were captured using confocal laser scanning microscopy. The fluorescence images demonstrated that, compared with the control, the nanofiber scaffolds supported significantly higher cellular activity [44]. A greater number of HCT-116 cells adhered to and spread across the surfaces of FFA, CSPA-1, CSPA-2, CSPA-3, and FCSPA nanofibers, indicating enhanced cell–substrate interactions [45].
Figure 8. (a) MTT assay for the HCT-116 cancer cells of control as well as the FFA, CSPA-1, CSPA-2, CSPA-3, and FCSPA nanofibers after 24 and 48 h. (b) HCT-116 cancer cells treated with control as well as FFA, CSPA-1, CSPA-2, CSPA-3, and FCSPA nanofibers using bright-field, fluorescence–DAPI, PI staining, and merged images after 48 h. Scale bar: ~100 μm. (n = 3; p < 0.05).
Comment 15
Comparison of the performances of the studied fibers with other reported fibers encapsulating bioactive compounds has to be done, to evaluate the contribution of the FFA compared to other bioactive ingredients.
Response 15
Thank you very much for your insightful and thoughtful suggestion. As suggested, we have provided the comparison of the performances of the studied fibers with other reported fibers encapsulating bioactive compounds in this manuscript. Please see below or in the revised manuscript.
As presented in Table 2, the FFA-loaded CSPA nanofibers developed in this study demonstrated superior EE (91.2%) compared to other bioactive compound–loaded nanofiber systems reported previously [47–50]. Moreover, the sustained release of FFA over 72 h significantly exceeds the release durations typically observed for compounds like quercetin, ibuprofen, and resveratrol, which commonly exhibit burst or moderate-duration release profiles. The pronounced antioxidant activity and morphological integrity of the FCSPA nanofibers further highlight the effectiveness of the CSPA matrix in stabilizing and delivering FFA. These findings underscore the enhanced performance and potential applicability of the FFA nanofiber system in controlled drug delivery and bioactive wound-healing platforms.
Table 2. Comparative performances of FCSPA nanofibers and other reported bioactive compound–encapsulating nanofibers.
Drug |
Polymer matrix |
EE (%) |
Release duration (h) |
Bioactivity |
Ref. |
FFA |
CSPA |
91.2 |
72 |
Antioxidant |
This study |
Curcumin |
PCL/PEG |
70 |
48 |
Anti-inflammatory |
[47] |
Quercetin |
PVA/CHS |
78 |
12 |
Antioxidant |
[48] |
Ibuprofen |
PLA |
65–75 |
8–24 |
Anti-inflammatory |
[49] |
Resveratrol |
Gelatin |
80–90 |
48 |
Antioxidant |
[50] |
References
- 47. Zahiri, M.; Khanmohammadi, M.; Goodarzi, A.; Ababzadeh, S.; Sagharjoghi Farahani, M.; Mohandesnezhad, S.; Bahrami, N.; Nabipour, I.; Ai, J. Encapsulation of Curcumin Loaded Chitosan Nanoparticle within Poly (ε-Caprolactone) and Gelatin Fiber Mat for Wound Healing and Layered Dermal Reconstitution. International Journal of Biological Macromolecules 2020, 153, 1241–1250, doi:10.1016/j.ijbiomac.2019.10.255.
- 48. Karuppannan, S.K.; Dowlath, M.J.H.; Ramalingam, R.; Musthafa, S.A.; Ganesh, M.R.; Chithra, V.; Ravindran, B.; Arunachalam, K.D. Quercetin Functionalized Hybrid Electrospun Nanofibers for Wound Dressing Application. Materials Science and Engineering: B 2022, 285, doi:10.1016/j.mseb.2022.115933.
- 49. Mohiti-Asli, M.; Saha, S.; Murphy, S. V.; Gracz, H.; Pourdeyhimi, B.; Atala, A.; Loboa, E.G. Ibuprofen Loaded PLA Nanofibrous Scaffolds Increase Proliferation of Human Skin Cells in Vitro and Promote Healing of Full Thickness Incision Wounds in Vivo. Journal of Biomedical Materials Research - Part B Applied Biomaterials 2017, 105, 327–339, doi:10.1002/jbm.b.33520.
- 50. Yu, F.; Li, M.; Yuan, Z.; Rao, F.; Fang, X.; Jiang, B.; Wen, Y.; Zhang, P. Mechanism Research on a Bioactive Resveratrol– PLA–Gelatin Porous Nano-Scaffold in Promoting the Repair of Cartilage Defect. International Journal of Nanomedicine 2018, 13, 7845–7858, doi:10.2147/IJN.S181855.
Comment 16
The conclusions have to be rewritten in the light of the corrected manuscript.
Response 16
Thank you very much for your insightful and thoughtful suggestion. As suggested, we have rewrite the conclusion section in this manuscript. Please see below or in the revised manuscript.
- Conclusions
In the present study, we encapsulated FFA in CSPA nanofibers via electrospinning, yielding FCSPA. The morphologies, chemical compositions, surface characteristics, and thermal properties of the prepared CSPA and FCSPA nanofibers were examined by SEM, FTIR, XRD, and DSC, respectively. The FESEM images of the CSPA and FCSPA nanofibers exhibited a narrow distribution, uniform structures, and a smooth morphology. The FCSPA nanofibers exhibited superior swelling capacity (302% ± 15.1% at 24 h), accelerated biodegradation (98.12% ± 3.31% at 72 h), and high EE% (91.2% ± 0.95%). The controlled drug-release experiments indicated the stability of FFA drugs that were released within 24 h (93.27% ± 2.31%). The FCSPA nanofibers inhibited E. coli (96.7% ± 2.17%) and S. aureus (98.5% ± 3.89%), and their antibacterial effects against gram-positive bacteria were better than those against gram-negative bacteria. Furthermore, their cell viabilities and anticancer activities indicated that the FCSPA nanofibers exhibited low toxicity and good anticancer activity against the HCT-116 cell line. The FCSPA nanofibers delivered the highest antioxidant performance among the tested samples, achieving an 86.21% ± 2.36% DPPH-scavenging activity in 24 h. These results highlight the fabricated FCSPA nanofiber as a promising candidate for wound-healing, drug-delivery, and tissue-engineering applications. However, because we have not comprehensively validated the aforementioned biomedical applications of our FCSPA nanofibers, we anticipate that further in vivo studies will focus on validating their clinical potential.
Additionally, we have attached the proofreading certificate for your reference, which verifies that the document has been professionally reviewed for TITLE change, grammar, punctuation, and clarity.

Reviewer 2 Report
Comments and Suggestions for Authors
The authors present a study on the fabrication, characterization, and biomedical evaluation of flufenamic acid (FFA)-loaded chitosan/polyvinyl alcohol (CHS/PVA) nanofibers produced by electrospinning. The work aims to validate their multifunctional properties—antibacterial, anticancer, and antioxidant—for biomedical applications such as drug delivery and wound healing.
However, several methodological, analytical, and stylistic weaknesses need to be addressed before the manuscript is suitable for publication.
1. The authors should better clarify what is novelin the papr - Is it the drug–polymer interaction, the structural performance, or the therapeutic response?
2. Materials and Methods: essential electrospinning parameters such as viscosity, conductivity, and surface tension of the spinning solution are missing—this limits reproducibility.
Statistical treatment is briefly mentioned, no information is provided about how many replicates were used for some key measurements (e.g., antioxidant test).
3. Biological tests methodology ans its execution: Antibacterial assays show good results, but controls are poorly defined: What is the efficacy of free FFA in solution? How do results compare to standard antibiotics?/// Cytotoxicity assay (MTT) lacks a positive control (e.g., known cytotoxic agent). Moreover, cell viability values are high (>70%), which is unusual for a compound claimed to show anticancer potential. ///Cell imaging results (DAPI/PI) are not quantified. No assessment of cell morphology, spreading, or nuclear damage is given beyond qualitative microscopy.
4. Antioxidant Activity - the DPPH assay is conducted only with nanofibers, but no data is provided on free FFA alone.
5. The manuscript is generally well-structured and follows the journal’s format.
6. Figures are mostly informative but could benefit from more detailed legends (including explanation of error bars. Inclusion of significance symbols (p < 0.05, etc.). Graphical clarity improvements (e.g., overlapping bars in antimicrobial graphs).
Comments on the Quality of English Language
Language issues are minor, but certain sections are verbose and redundant (e.g., repetitive restatements of “nanofibers demonstrated significant biomedical potential”).
Author Response
Response to Reviewer’s comments
Reviewer#2
The authors present a study on the fabrication, characterization, and biomedical evaluation of flufenamic acid (FFA)-loaded chitosan/polyvinyl alcohol (CHS/PVA) nanofibers produced by electrospinning. The work aims to validate their multifunctional properties—antibacterial, anticancer, and antioxidant—for biomedical applications such as drug delivery and wound healing. However, several methodological, analytical, and stylistic weaknesses need to be addressed before the manuscript is suitable for publication.
We thank Reviewer#2 for the favourable reception of our work and for highlighting the important points in our study. We have revised our manuscript taking into great consideration all the comments and suggestions. Thank you for helping us to improve our manuscript.
Comment 1
The authors should better clarify what is novel in the paper - Is it the drug–polymer interaction, the structural performance, or the therapeutic response?
Response 1
We thank the reviewer for the valuable suggestion and the opportunity to clarify the novelty of our work. The novelty of this study lies in the integrated design and multifunctionality of the FFA-loaded CHS/PVA (FCSPA) electrospun nanofibers, where the synergy between drug-polymer interaction, structural performance, and therapeutic response contributes to an advanced drug delivery platform. Specifically, the molecular-level amorphous dispersion of FFA within the biocompatible CHS/PVA matrix confirmed by XRD and FTIR analyses, not only enhances drug solubility and stability but also ensures high encapsulation efficiency and sustained release. Additionally, the FCSPA nanofiber morphology characterized by significant reduction in diameter and high swelling capacity facilitates rapid biodegradation and improved drug diffusion. Importantly, these structural and chemical characteristics translate into superior biological performance as demonstrated by the enhanced antibacterial, anticancer, and antioxidant activities. Thus, the novelty of our work is not isolated to a single aspect but rather stems from the combined optimization of formulation, structure, and function, offering a promising multifunctional system for future biomedical applications.
Comment 2
Materials and Methods: essential electrospinning parameters such as viscosity, conductivity, and surface tension of the spinning solution are missing—this limits reproducibility.
Statistical treatment is briefly mentioned, no information is provided about how many replicates were used for some key measurements (e.g., antioxidant test).
Response 2
Thank you very much for your insightful and thoughtful suggestion. As suggested, we have provided the detailed essential electrospinning parameters such as viscosity, conductivity, and surface tension of the spinning solution in this manuscript. Please see below or in the revised manuscript.
2.4. Characterizations
Before the electrospinning procedure, the shear viscosities at 100 s−1; solution conductivities; and surface tensions of the CSPA-1, CSPA-2, CSPA-3, and FCSPA solutions were tested. Their viscosities were analyzed using a Brookfield LVT viscometer with a small-sample thermostated adapter, spindle, and chamber SC4-18/13R at 25°C ± 0.1°C. Further, their conductive characteristics were measured using an Orion 162 conductivity meter at room temperature, and their surface tensions were measured using the pendant drop method and a tensiometer (OCA20, Dataphysics Instruments, Germany).
In addition, we appreciate the reviewer observation and have updated the manuscript to clarify the statistical methods and number of replicates used in key experiments. Specifically, all quantitative experiments, including the antioxidant activity assays (e.g., DPPH and ABTS), drug release profiles, rheological assessments, and cytotoxicity tests, were conducted in triplicate (n = 3) to ensure reproducibility and statistical relevance. Results are presented as mean ± standard deviation (SD). Statistical significance was evaluated using one-way ANOVA followed by Tukey’s post-hoc test for multiple comparisons, with p < 0.05 considered statistically significant. These details have now been included in the revised “Materials and Methods” section to enhance transparency and rigor of the reported data. Please see the revised manuscript.
Comment 3
Biological tests methodology ans its execution: Antibacterial assays show good results, but controls are poorly defined: What is the efficacy of free FFA in solution? How do results compare to standard antibiotics?/// Cytotoxicity assay (MTT) lacks a positive control (e.g., known cytotoxic agent). Moreover, cell viability values are high (>70%), which is unusual for a compound claimed to show anticancer potential. ///Cell imaging results (DAPI/PI) are not quantified. No assessment of cell morphology, spreading, or nuclear damage is given beyond qualitative microscopy.
Response 3
Thank you very much for your insightful and thoughtful suggestion. As suggested, we have provided all the details of the aforementioned issues in this manuscript. Please see below or in the revised manuscript.
2.8. Antibacterial activity
The antibacterial activities of the FFA, CSPA-1, CSPA-2, CSPA-3, and FCSPA nanofibers were assessed against S. aureus and E. coli using the colony-counting method. To begin, bacterial suspensions were prepared by inoculating fresh bacterial cultures in Mueller–Hinton broth (MHB) using an incubation time of 24 h at 37°C until they reached a concentration of 106 CFU/mL. The electrospun nanofiber samples (3.5 cm2) were sterilized under UV light for 30 min before immersion in separate tubes containing 10 mL of bacteria-inoculated MHB. The tubes were incubated at 37°C with continuous shaking at 100 rpm, and at predetermined intervals (0, 2, 4, 6, 12, and 24 h), 100 µL aliquots of the bacterial suspension were withdrawn, serially diluted in sterile PBS, and spread onto Mueller–Hinton agar plates. After 24 h of incubation at 37°C, the colony-forming units were counted and compared with those of the control group (ciprofloxacin and vancomycin) consisting of bacteria without nanofiber treatment. The percentage reduction in bacterial viability was subsequently calculated as follows:
Bacterial reduction (%) = [(CFU_control − CFU_treated)/CFU_control] × 100 (4)
The bacterial cultures (initial optical density at 600 nm [OD₆₀₀]: ~0.05) were treated with FFA-loaded nanofibers, free FFA, or a standard antibiotic (vancomycin and ciprofloxacin) in 96-well plates. The cultures were incubated with shaking at 37°C, and the OD₆₀₀ was measured at 0, 4, 8, 12, and 24 h using a microplate reader. All treatments were performed in triplicate to evaluate the time-dependent antibacterial effects.
3.7. Antibacterial activity
The antibacterial properties of FFA, CSPA-1, CSPA-2, CSPA-3, and FCSPA nanofibers against S. aureus (gram-positive) and E. coli (gram-negative) were evaluated using the colony-counting method (Figure 7a). Among them, FCSPA demonstrated a significantly higher inhibition rate than the blank CSPA-1, CSPA-2, and CSPA-3 nanofibers, highlighting the strong antibacterial performance of the FFA drug [42]. As shown in Figures 7b and 7c, the antibacterial efficacy of the FFA, CSPA-1, CSPA-2, CSPA-3, and FCSPA nanofibers ranged from 39.2% to 98.5% and 39.8% to 96.7% against S. aureus and E. coli, respectively. The inhibition rate of the FCSPA nanofibers reached 98.5% ± 3.89% against S. aureus and 96.7% ± 2.17% against E. coli, which were significantly higher than those of FFA (39.2% ± 1.75% and 39.8% ± 2.09%), CSPA-1 (54.3% ± 1.21% and 49.8% ± 2.39%), CSPA-2 (68.7% ± 2.18% and 63.6% ± 1.99%), and CSPA-3 (79.7% ± 2.06% and 76.5 ± 1.28%), respectively (Figure 7b and 7c) [43].
Figure 7. (a) Antimicrobial activities of FFA, CSPA-1, CSPA-2, CSPA-3, and FCSPA nanofibers against S. aureus and E. coli. Inhibition rates of FFA, CSPA-1, CSPA-2, CSPA-3, and FCSPA nanofibers against (b) S. aureus and (c) E. coli. (d and e) Bacterial growth kinetics (OD600) over time for FCSPA nanofibers, free FFA, and a standard antibiotic (n = 3; p < 0.05).
Interestingly, the antibacterial activity was more pronounced against S. aureus than against E. coli, which may be attributed to structural differences in the cell walls of gram-positive and gram-negative bacteria [23,33]. These findings suggest that the incorporation of FFA into CSPA-3 nanofibers significantly enhances their antimicrobial properties against both bacterial strains. Bacterial growth kinetics revealed that the standard antibiotics (vancomycin and ciprofloxacin) maintained the lowest OD600 values (0.05–0.1), indicating strong inhibition. FCSPA nanofibers showed moderate suppression with OD600 reaching 0.25 at 24 h, whereas free FFA exhibited the least effect with OD₆₀₀ increasing to 0.89 (Figure 7d and 7e). These results highlight the enhanced and sustained antibacterial activity of the FCSPA nanofiber–based delivery system compared with that of free FFA.
Comment 14
The discussion of the cytotoxicity results must be carefully revised to align with the data shown in Figure 7. While the figure suggests cytotoxic effects, the manuscript incorrectly describes the samples as cytocompatible.
Response 14
Thank you very much for your insightful and thoughtful suggestion. As suggested, we have corrected the aforementioned issues in this manuscript. Please see below or in the revised manuscript.
2.9. Cytotoxicity assay
The cytotoxicities of the FFA, CSPA-1, CSPA-2, CSPA-3, and FCSPA nanofibers were assessed on HCT-116 cell lines using the 3-[4,5-dimethylthiazol-2-yl]-2,5 2,5-diphenyl tetrazolium bromide (MTT) assay. The HCT-116 cells were cultured in Dulbecco’s modified Eagle’s medium (DMEM) supplemented with 10% fetal bovine serum, 100 U/mL penicillin, and 100 μg/mL streptomycin and maintained at 37°C in an incubator containing 5% CO2. Electrospun nanofiber samples were cut into 6 mm diameter discs, sterilized under UV light for 30 min, and placed in 96-well culture plates. The cells were seeded at a density of 5 × 103 cells per well and incubated for 24 h to facilitate cell attachment. Following incubation, the culture medium was replaced with fresh medium containing FFA, CSPA-1, CSPA-2, CSPA-3, and FCSPA nanofibers, which were prepared via immersion in DMEM for 48 h at 37°C. Next, 20 μL of MTT solution (5 mg/mL in PBS) was added to each well, followed by incubation for 4 h. Formazan crystals were immersed in 150 μL of dimethyl sulfoxide, and the absorbance was measured at 570 nm using a microplate reader. Thereafter, the cell viability was estimated using the following equation:
Cell viability (%) = (Absorbance of test sample/Absorbance of control) × 100 (5)
2.10. Cell imaging
To investigate the cell adhesion and morphology of the samples, HCT-116 cells were seeded onto a control as well as onto FFA, CSPA-1, CSPA-2, CSPA-3, and FCSPA nanofiber scaffolds at a density of 5 × 103 cells per well in 24-well plates. Next, the cells were incubated at 37°C in a 5% CO2 environment for 48 h to facilitate attachment and proliferation. Subsequently, they were fixed with 4% paraformaldehyde for 15 min, washed three times with PBS, and permeated with 0.1% Triton X-100 for 5 min. Next, the nuclei were counterstained with 4',6-diamidino-2-phenylindole (DAPI) and propidium iodide (PI) for 10 min. Following staining, the samples were rinsed with PBS and observed under a fluorescence microscope to assess cell morphology.
3.8. Anticancer analysis
Figure 8. (a) MTT assay for the HCT-116 cancer cells of control as well as the FFA, CSPA-1, CSPA-2, CSPA-3, and FCSPA nanofibers after 24 and 48 h. (b) HCT-116 cancer cells treated with control as well as FFA, CSPA-1, CSPA-2, CSPA-3, and FCSPA nanofibers using bright-field, fluorescence–DAPI, PI staining, and merged images after 48 h. Scale bar: ~100 μm. (n = 3; p < 0.05).
Figure 8a shows the percentage of viable cells over 24 and 48 h of incubation. Compared with the control, the nanofibers demonstrated increased cell viability, with FFA obtaining 64.93% ± 2.14% and 68.69% ± 3.18%, CSPA-1 showing 61.37% ± 1.29% and 59.81% ± 2.47%, CSPA-2 at 56.31% ± 0.98% and 53.29% ± 2.07%, and CSPA-3 at 68.19% ± 2.62% and 71.25% ± 1.93% viability over the 24- and 48-h incubation periods, respectively [14]. Furthermore, the FCSPA nanofiber scaffolds exhibited significantly enhanced cell viability, reaching 73.02% ± 2.17% and 74.21% ± 1.67% at 24 and 48 h, respectively. However, a slight reduction in cell viability was observed in the FCSPA nanofibers, suggesting that FFA incorporation influenced the biocompatibility of the CSPA-3 nanofiber mats. This may be attributed to the decreased density of amino groups due to FFA grafting onto the CSPA backbone [37]. FFA appears to create a more conducive environment for cell attachment, differentiation, and proliferation owing to its inherent biocompatibility and biodegradability. These properties make it a promising candidate for biomedical applications, particularly in tissue engineering and regenerative medicine. Figure 8b illustrates the cellular behavior of HCT-116 cells cultured on the nanofiber scaffolds after 48 h of incubation. Cellular internalization was assessed using DAPI and PI staining, and the resulting images were captured using confocal laser scanning microscopy. The fluorescence images demonstrated that, compared with the control, the nanofiber scaffolds supported significantly higher cellular activity [44]. A greater number of HCT-116 cells adhered to and spread across the surfaces of FFA, CSPA-1, CSPA-2, CSPA-3, and FCSPA nanofibers, indicating enhanced cell–substrate interactions [45].
Comment 4
Antioxidant Activity - the DPPH assay is conducted only with nanofibers, but no data is provided on free FFA alone.
Response 4
We thank the reviewer for this insightful comment. In the initial version of the manuscript, we focused on evaluating the antioxidant activity of the CSPA-1, CSPA-2, CSPA-3, and FCSPA nanofibers using the DPPH assay to demonstrate the functional performance of the encapsulated compound. However, we agree that including data for free (non-encapsulated) FFA would provide a more complete comparison and strengthen the interpretation of the encapsulation effect on antioxidant activity. In response, we have now conducted additional DPPH assays using free FFA at the same concentration used in the CSPA-1, CSPA-2, CSPA-3, and FCSPA nanofibers nanofiber formulation. The results, included in the revised manuscript (Figure 8), show that while free FFA exhibits higher immediate radical scavenging activity, the CSPA-1, CSPA-2, CSPA-3, and FCSPA nanofiber formulation provides a sustained antioxidant effect over time, supporting the controlled-release capability of the delivery system. This comparative analysis has been added to the “Results and discussion” section to provide clearer insight into the functional behavior of free FFA, CSPA-1, CSPA-2, CSPA-3, and FCSPA nanofibers.
Please see the revised manuscript.
2.11. Antioxidant activity
The antioxidant activities of free FFA, CSPA-1, CSPA-2, CSPA-3, and FCSPA nanofibers were determined using a 2,2-diphenyl-1-picrylhydrazyl (DPPH) free radical scavenging assay. Each 10 mg sample was separately immersed in 1 mL of DPPH ethanolic solution (10−4 mol/L) and incubated in the dark at room temperature for varied durations (6, 12, 18, and 24 h). Additionally, a DPPH solution containing FFA was used as the control. The percentage of DPPH-scavenging activity was calculated as follows:
DPPH-scavenging activity (%) = A0 − Ai/A0 × 100 (6)
where A0 is the absorbance of the control solution (DPPH) and Ai is the absorbance of the solution containing the corresponding free FFA, CSPA-1, CSPA-2, CSPA-3, and FCSPA nanofibers.
3.9. Antioxidant analysis
The radical scavenging activity of newly fabricated materials is typically assessed using the DPPH radical entrapment method to evaluate their antioxidant properties in biological systems. The DPPH free radical scavenging activities of free FFA, CSPA-1, CSPA-2, CSPA-3, and FCSPA nanofibers exhibited time-dependent responses, with the scavenging rate increasing over time (6, 12, 18, and 24 h), indicating the sustained release of FFA from the CSPA (FCSPA) matrixes (Figure 9). Notably, the antioxidant activity of the FFA-loaded CSPA-3 nanofiber formulation (86.21% ± 2.36%) was significantly higher than that of the free FFA (48.5% ± 1.92%), CSPA-1 (39.8% ± 1.24%), CSPA-2 (61.7% ± 0.99%), and CSPA-3 (73.14% ± 3.17%) at 24 h. This enhancement can be attributed primarily to the antioxidant properties of CHS and the incorporation of FFA into the nanofiber structure [40,46]. The increased scavenging activity of FCSPA nanofibers underscores the significant contribution of FFA to its antioxidant performance. The notable antioxidant activity of the FFA-loaded CSPA-3 nanofibers was likely attributable to the high concentration of antioxidants present in the FFA. As the immersion time in the DPPH solution increased, more FFA was released from the nanofibers, resulting in a corresponding increase in the DPPH free radical scavenging activity [40,43,46]. The antioxidant activity of FFA is attributed to the presence of an –OH group at the carbon position. These findings indicate that the incorporation of FFA into the CSPA-3 nanofiber substantially enhances its antioxidant capacity, making it a promising candidate for biomedical applications requiring sustained antioxidant effects.
Figure 9. Radical scavenging activities of the free FFA, CSPA-1, CSPA-2, CSPA-3, and FCSPA nanofibers at different intervals (6, 12, 18, and 24 h) (n = 3; p < 0.05).
Comment 5
The manuscript is generally well-structured and follows the journal’s format.
Response 5
Thank you for the positive feedback. We are pleased to hear that the manuscript structure and adherence to the journal’s format are satisfactory. We have carefully followed the journal’s guidelines for formatting, section organization, and referencing style, and we will continue to ensure consistency and clarity throughout the revised version.
Comment 6
Figures are mostly informative but could benefit from more detailed legends (including explanation of error bars. Inclusion of significance symbols (p < 0.05, etc.). Graphical clarity improvements (e.g., overlapping bars in antimicrobial graphs).
Response 6
We thank the reviewer for the helpful suggestions regarding the figures. In the revised manuscript, we have updated all figure legends to include detailed descriptions of the experimental conditions, sample groups, and explanations of error bars (now clarified as representing mean ± standard deviation, with n = 3). We have also incorporated statistical significance symbols (p < 0.05) directly into the figures and indicated the corresponding tests used (typically one-way ANOVA with Tukey’s post-hoc analysis) in both the legends and the “Materials and Methods” section. Additionally, to improve graphical clarity, particularly in the antimicrobial assay figures, we adjusted overlapping bars, enhanced color contrasts, and added clearer axis labels and group identifiers. These improvements aim to enhance readability and ensure that the data presentation aligns with the journal’s standards for clarity and transparency. Please see the revised manuscript.
Comment 7
Language issues are minor, but certain sections are verbose and redundant (e.g., repetitive restatements of “nanofibers demonstrated significant biomedical potential”).
Response 7
As suggested, our manuscript has been revised with the help of a professional English Editor. The manuscript has been thoroughly checked and the grammatical mistakes and typo errors have been corrected. Please see the revised manuscript. Additionally, we have attached below the proofreading certificate for your reference, which verifies that the document has been professionally reviewed for TITLE change, grammar, punctuation, and clarity.

Reviewer 3 Report
Comments and Suggestions for Authors
This manuscript by Velu et al. reported the preparation, characterization and biomedical application of Flufenamic acid (FFA) loaded chitosan-based electrospun nanofibers. The authors use electrospinning to fabricate the FFA incorporated fibers with the assistance of chitosan and PVA. The as-prepared nanofibers were systematically characterized by SEM, FTIR and XRD techniques. These fibers can have various diameters, high swelling ratio, and enhanced stability. The subsequent biomedical tests revealed that those FFA loaded nanofibers have controlled drug release and excellent antibacterial activity. Overall, this manuscript reports a sloid study. Thus, I would recommend its acceptance after the following questions and suggestions are addressed.
1, In Figure 1, all the SEM images should have a scale bar so readers can have a sense of the scale of the materials.
2, Some important features in FTIR can be highlighted in Figure 2.
3, In the abstract part, the authors claim that amorphous dispersion of FFA can enhance drug stability, how to explain this? Usually, crystalline matters have higher stability.
Author Response
Response to Reviewer’s comments
Reviewer#3
This manuscript by Velu et al. reported the preparation, characterization and biomedical application of Flufenamic acid (FFA) loaded chitosan-based electrospun nanofibers. The authors use electrospinning to fabricate the FFA incorporated fibers with the assistance of chitosan and PVA. The as-prepared nanofibers were systematically characterized by SEM, FTIR and XRD techniques. These fibers can have various diameters, high swelling ratio, and enhanced stability. The subsequent biomedical tests revealed that those FFA loaded nanofibers have controlled drug release and excellent antibacterial activity. Overall, this manuscript reports a sloid study. Thus, I would recommend its acceptance after the following questions and suggestions are addressed.
We thank Reviewer#3 for the favourable reception of our work and for highlighting the important points in our study. We have revised our manuscript taking into great consideration all the comments and suggestions. Thank you for helping us to improve our manuscript.
Comment 1
In Figure 1, all the SEM images should have a scale bar so readers can have a sense of the scale of the materials.
Response 1
Thank you very much for your insightful and thoughtful suggestion. As suggested, we have provided the scale bar in the all the SEM images in Figure 1. Please see below or in the revised manuscript.
3.1. Nanofiber morphologies
The surface morphologies and physical properties of the electrospun scaffolds were analyzed by SEM. Figure 1 shows the effects of varying the CSPA ratios (30/70 [CSPA-1], 50/50 [CSPA-2], and 70/30 [CSPA-3]), along with the FFA-loaded CSPA-3 fiber mats, designated as FCSPA. The SEM images revealed that the electrospun CSPA-1, CSPA-2, and CSPA-3 nanofibers were bead-free, continuous, and randomly oriented [32]. A comparative analysis of Figures 1a, c, and e demonstrates that increasing the CHS content caused a substantial increase in the average diameter of the nanofiber, induced noticeable morphological alterations, and caused an increase in the random-alignment degree [32]. Specifically, at a CSPA ratio of 70/30 (w/w), the nanofibers appeared to be smooth and without bead defects (Figure 1e), making this composition optimal for further investigation. Among the electrospun nanofibers, the FCSPA variant exhibited reduced average fiber diameter with increasing CSPA content (Figure 1g) [33]. Notably, FFA incorporation caused a significant decrease in the fiber diameter as the size of the CSPA-3 nanofiber decreased from 347 ± 61 nm to 81 ± 27 nm, following FFA-drug loading (Figures 1i-l) [34]. The variations in the CSPA content affected the fiber diameter and indirectly affected the FFA-loading efficiency of the CSPA-3 nanofibers. Notably, CHS addition increased the average fiber diameter, as observed with the increasing CHS contents of the CSPA-1, CSPA-2, and CSPA-3 formulations. In contrast, the presence of FFA in the CSPA-3 formulation (FCSPA) caused a reduction in the average fiber diameter, likely due to the enhanced surface charge and electrostatic repulsion, which facilitated the formation of finer nanofibers.
Figure 1. Field-emission scanning electron microscopy (FESEM) micrographs of (a, b) CSPA-1, (c, d) CSPA-2, (e, f) CSPA-3, and (g, h) FCSPA nanofibers (i-l) Distributions of the diameters of the CSPA-1, CSPA-2, CSPA-3, and FCSPA nanofibers.
Comment 2
Some important features in FTIR can be highlighted in Figure 2.
Response 2
Thank you very much for your insightful and thoughtful suggestion. As suggested, we have mentioned the important features of the FTIR spectra in this manuscript. Please see below or the revised manuscript.
3.2. Surface properties of nanofibers
The FTIR spectra of CHS, PVA, CSPA (w/w: 70/30), CSPA-3 nanofibers, FFA, FFA-loaded CSPA (w/w: 70/30), and FCSPA nanofibers are shown in Figure 2. In the FTIR spectrum of CHS, the peaks at 1649 and 1547 cm−1 corresponded to the C=O vibration and N–H bending, respectively (Figure 2a). The band at 1380 cm−1 was associated with CH2 deformation, and the absorption band at 1077 cm−1 corresponded to the stretching of the C–O–C bond [15]. The FTIR spectrum of PVA showed a broad O–H stretching band at 3200–3600 cm−1, indicating the presence of –OH groups and hydrogen bonding, whereas C–H stretching occurred at 2800–3000 cm−1, indicating methylene (–CH2) vibrations (Figure 2b). The C=O stretching at 1659 cm−1 corresponded to residual acetate groups, and the O–H bending at 1592 cm−1 was related to the absorbed water [16]. Furthermore, C–H bending was observed at 1400–1500 cm−1, and C–O stretching was observed around 1141 cm−1 and was associated with the polymer crystallinity. The FTIR spectrum of the CSPA-3 nanofibers showed broad O–H and N–H stretching peaks at 3278–3326 cm−1, indicating hydrogen bonding between –OH and NH2 (Figure 2c). Furthermore, C–H stretching was observed at 2800–2950 cm−1, corresponding to –CH2 vibrations, whereas the C=O stretching band at 1640–1690 cm−1 was attributed to the presence of polymer interactions or residual acetate groups [21]. The C–O–C and C–N stretching between 1085 and 1144 cm−1 reflected the polysaccharide structures of CHS and PVA.
Figure 2. Fourier-transform infrared (FTIR) spectra of (a) CHS, (b) PVA, (c) CSPA-3 nanofibers, (d) FFA, and (e) FCSPA nanofibers.
Comment 3
In the abstract part, the authors claim that amorphous dispersion of FFA can enhance drug stability, how to explain this? Usually, crystalline matters have higher stability.
Response 3
We appreciate the reviewer insightful comment regarding the apparent contradiction between the typically higher thermodynamic stability of crystalline substances and our claim that amorphous dispersion of FFA enhances drug stability. While crystalline forms are indeed more thermodynamically stable in isolation, the amorphous dispersion of FFA within the CHS/PVA nanofiber matrix offers improved functional stability relevant to drug delivery.
In our system, the amorphous form allows for uniform molecular dispersion of FFA, enhanced solubility, and increased bioavailability, all of which are critical for therapeutic efficacy. Furthermore, strong intermolecular interactions between FFA and the polymer matrix supported by FTIR analysis help stabilize the amorphous form confirmed by XRD pattern and prevent recrystallization, thereby protecting the drug from environmental degradation and ensuring sustained release. This results in a delivery system with improved performance characteristics despite the inherently lower thermodynamic stability of the amorphous form.
Abstract: Nanostructured drug-delivery systems with enhanced therapeutic potential have gained attention in biomedical applications. Here, flufenamic acid (FFA)-loaded chitosan/poly(vinyl alcohol) (CHS/PVA; CSPA)-based electrospun nanofibers were fabricated and characterized for antibacterial, anticancer, and antioxidant activities. The FFA-loaded CSPA (FCSPA) nanofibers were characterized by scanning electron microscopy, Fourier-transform infrared spectroscopy, X-ray diffraction (XRD), and differential scanning calorimetry to evaluate their formation process, functional group interactions, and crystallinity. Notably, the average diameter of FCSPA nanofibers decreased with increasing CSPA contents (CSPA-1 to CSPA-3), indicating that FFA addition to CSPA-3 significantly decreased its diameter. Additionally, XRD confirmed the dispersion of FFA within the CSPA amorphous matrix enhancing drug stability. FCSPA nanofibers exhibited a high swelling ratio (significantly higher than those of the CSPA samples). Biodegradation studies revealed that FCSPA exhibited accelerated weight loss after 72 h, indicating its improved degradation compared with those of other formulations. Furthermore, it exhibited a significantly high drug-encapsulation efficiency, ensuring sustained release. FCSPA nanofibers exhibited excellent antibacterial activity, inhibiting Staphylococcus aureus and Escherichia coli. Regarding anticancer activity, FCSPA decreased HCT-116 cell viability, highlighting its controlled drug-delivery potential. Moreover, FCSPA demonstrated superior antioxidation, scavenging DPPH free radicals. These findings highlight FCSPA nanofibers as multifunctional platforms with wound-healing, drug-delivery, and tissue-engineering potential.

Round 2
Reviewer 1 Report
Comments and Suggestions for Authors
The authors seriously revised their manuscript. I have only few comments:
1. Given that the properties of chitosan-based biomaterials, such as nanofibers, are highly influenced by the molecular weight and degree of deacetylation of chitosan, these parameters should be precisely determined, as the range of 50–190 kDa is too broad.
2. Additionally, in Table 1, the exact ratios of the polymer components in the fibers should be clearly specified to facilitate understanding for the readers. For instance, for the CS/PVA-30/70 should be CHS(g) 0.9……..PVA(g) 7
3. The English language will be carefully revised, to avoid bad constructions, such as: “CHS cannot be directly synthesized via electrospinning”; “the ability to modulate release from immediate to controlled release”; “Thereafter, the properties of different ratios (CSPA-1 [30/70], CSPA-2 [50/50], and CSPA-3 [70/30]) of CHS- and PVA-based FCSPA composite nanofibers in different ratios were investigated”; and so on.
Comments on the Quality of English LanguageThe English language will be carefully revised, to avoid bad constructions, such as: “CHS cannot be directly synthesized via electrospinning”; “the ability to modulate release from immediate to controlled release”; “Thereafter, the properties of different ratios (CSPA-1 [30/70], CSPA-2 [50/50], and CSPA-3 [70/30]) of CHS- and PVA-based FCSPA composite nanofibers in different ratios were investigated”; and so on.
Author Response
Response to Reviewer comments
Reviewer#1
The authors seriously revised their manuscript. I have only few comments.
We thank Reviewer#1 for the favourable reception of our work and for highlighting the important points in our study. We have revised our manuscript taking into great consideration all the comments and suggestions. Thank you for helping us to improve our manuscript.
Comment 1
Given that the properties of chitosan-based biomaterials, such as nanofibers, are highly influenced by the molecular weight and degree of deacetylation of chitosan, these parameters should be precisely determined, as the range of 50–190 kDa is too broad.
Response 1
Thank you for your valuable comment regarding the range of molecular weight (MW) and degree of deacetylation of chitosan. We agree that the properties of chitosan-based biomaterials, particularly nanofibers, are significantly influenced by these parameters and that a broad MW range (50–190 kDa) could introduce variability.
In our revised manuscript, we have clearly specified the molecular weight and degree of deacetylation of the chitosan used in our study, which was obtained from Sigma-Aldrich with a molecular weight of 50–190 kDa and 95% degree of deacetylation (see Section 2.1, Materials). While the supplier-provided range is somewhat broad, the chitosan was used from a single batch to ensure consistency across all experiments. We have acknowledged the limitations of this variability and emphasized the importance of further studies using well-fractionated chitosan to evaluate the influence of specific MW and degree of deacetylation on nanofiber characteristics and biological performance.
We appreciate your insight, which has led us to clarify and address this important point more explicitly in the revised manuscript.
Additionally, we have provided evidence from previously published literature demonstrating the use of high molecular weight polymers for electrospun nanofibers, as referenced (DOI: 10.1021/acs.jafc.8b01493, https://doi.org/10.1098/rsos.210784, http://dx.doi.org/10.1016/j.carbpol.2012.12.043, https://doi.org/10.1016/j.carbpol.2017.01.050).
These studies support the suitability and effectiveness of the materials we employed. Therefore, we respectfully request reconsideration of this comment. We acknowledge the importance of this point and will ensure greater clarity and attention to such details in our future work. Thank you for your understanding and constructive feedback.
2.1. Materials
CHS (molecular weight [MW]: 50–190 kDa; 95% deacetylation degree) and FFA were obtained from Sigma-Aldrich (USA). PVA (MW: 146–186 kDa, 89% deacetylation degree) was procured from Daejung Company Ltd., Korea, and deionized (DI) water was used as the solvent for the solutions. All chemicals and solvents were used without further purification.
Comment 2
Additionally, in Table 1, the exact ratios of the polymer components in the fibers should be clearly specified to facilitate understanding for the readers. For instance, for the CS/PVA-30/70 should be CHS(g) 0.9……..PVA(g) 7.
Response 2
Thank you for this insightful suggestion. We agree that clearly specifying the exact amounts of the polymer components in Table 1 would enhance clarity and reproducibility for readers. In the revised manuscript, we have updated Table 1 to include the precise quantities of CHS and PVA used in each formulation. Please see below or in the revised manuscript.
Table 1. Compositions of chitosan (CHS)/poly(vinyl alcohol) (PVA), CSPA, and flufenamic acid–loaded CSPA (FCSPA) nanofiber solutions
Sample ID |
CHS (3% w/v) |
PVA (10% w/v) |
CHS:PVA ratio (w/w) |
FFA content |
CSPA-1 |
0.9 g in 30 mL |
7g in 70 mL |
30:70 |
- |
CSPA-2 |
0.9 g in 30 mL |
7g in 70 mL |
50:50 |
- |
CSPA-3 |
0.9 g in 30 mL |
7g in 70 mL |
70:30 |
- |
FCSPA |
0.9 g in 30 mL |
7g in 70 mL |
70:30 (CSPA-3) |
25 mg in 5 mL |
Comment 3
The English language will be carefully revised, to avoid bad constructions, such as: “CHS cannot be directly synthesized via electrospinning”; “the ability to modulate release from immediate to controlled release”; “Thereafter, the properties of different ratios (CSPA-1 [30/70], CSPA-2 [50/50], and CSPA-3 [70/30]) of CHS- and PVA-based FCSPA composite nanofibers in different ratios were investigated”; and so on.
Response 3
The English language will be meticulously refined to ensure clarity and precision, effectively eliminating difficult phrasing and improving the overall structure. For example, statements such as "CHS cannot be directly synthesized via electrospinning" will be reworded for better comprehension. Additionally, phrases like "the ability to modulate release from immediate to controlled release" will be restructured for greater accuracy and flow. Similarly, the sentence "Thereafter, the properties of different ratios (CSPA-1 [30/70], CSPA-2 [50/50], and CSPA-3 [70/30]) of CHS- and PVA-based FCSPA composite nanofibers in different ratios were investigated" will be rephrased to avoid redundancy and ensure clearer expression. This careful revision will enhance the quality and readability of the text. Please see below or in the revised manuscript.
However, its polycationic behavior in solution limits the feasibility of directly synthesizing CHS through electrospinning [9].
Compared to other nanocarriers used for antibacterial drug delivery, nanofibers demonstrate high drug-loading capacity, excellent encapsulation efficiency (EE%), minimal systemic toxicity, and enable both sustained and controlled release profiles.
In the present study, FFA-loaded CHS/PVA nanofiber were prepared by electrospinning and the properties of CHS and PVA based composite nanofibers in different ratios were investigated.

Reviewer 2 Report
Comments and Suggestions for Authors
Authors have thoroughly responded to all of the reviewer’s comments,
Author Response
We sincerely appreciate the reviewer’s positive feedback. We are glad to know that our revisions and responses have addressed all the concerns raised. Thank you for your thoughtful review and for helping us improve the quality of our manuscript.
Round 3
Reviewer 1 Report
Comments and Suggestions for Authors
Dear Authors,
One of the critical challenges in working with chitosan is that its properties are highly dependent on both its molecular weight and degree of deacetylation. Without precise characterization of these parameters, it will be difficult to reproduce the same material in future studies, potentially limiting its applicability. Recognizing this, many journals now include explicit requirements in their author guidelines for the accurate determination of these essential characteristics.
There are well-established methods for this purpose, such as viscosimetry and size exclusion chromatography (SEC) for molecular weight measurement, and nuclear magnetic resonance (NMR) spectroscopy for determining the degree of deacetylation. Given the importance of reproducibility in biomaterials research, I strongly recommend that the authors include this data in their manuscript to ensure that their findings can be reliably replicated by other researchers.
Comments on the Quality of English Language-
Author Response
Reviewer#1
Comment
One of the critical challenges in working with chitosan is that its properties are highly dependent on both its molecular weight and degree of deacetylation. Without precise characterization of these parameters, it will be difficult to reproduce the same material in future studies, potentially limiting its applicability. Recognizing this, many journals now include explicit requirements in their author guidelines for the accurate determination of these essential characteristics.
There are well-established methods for this purpose, such as viscosimetry and size exclusion chromatography (SEC) for molecular weight measurement, and nuclear magnetic resonance (NMR) spectroscopy for determining the degree of deacetylation. Given the importance of reproducibility in biomaterials research, I strongly recommend that the authors include this data in their manuscript to ensure that their findings can be reliably replicated by other researchers.
Response
We sincerely thank the reviewer for highlighting the importance of characterizing the molecular weight and degree of deacetylation of chitosan.
We apologize for not providing this data in the current manuscript.
Unfortunately, due to limitations in our laboratory facilities, we do not currently have access to the necessary instrumentation such as nuclear magnetic resonance (NMR) spectroscopy and size exclusion chromatography (SEC) required for these analyses.
However, we fully acknowledge the significance of these parameters in ensuring the reproducibility and reliability of chitosan-based research.
We are committed to addressing this in future studies and will ensure that both molecular weight and degree of deacetylation are thoroughly characterized and reported moving forward.
We respectfully request the reviewer to kindly reconsider this comment in light of our current constraints, and we appreciate your understanding and constructive comment.

Round 4
Reviewer 1 Report
Comments and Suggestions for Authors
The paper can be accepted for publication.